# Structure of the lysosomal mTORC1–TFEB–Rag–Ragulator megacomplex

Zhicheng Cui[1,2], Gennaro Napolitano[3,4], Mariana E. G. de Araujo[5], Alessandra Esposito[3,4], Jlenia Monfregola[3,4], Lukas A. Huber[5], Andrea Ballabio[3,4,6,7,8 ✉] & James H. Hurley[1,2,9 ✉]

The transcription factor TFEB is a master regulator of lysosomal biogenesis and autophagy[1]. The phosphorylation of TFEB by the mechanistic target of rapamycin complex 1 (mTORC1)[2–5] is unique in its mTORC1 substrate recruitment mechanism, which is strictly dependent on the amino acid-mediated activation of the RagC GTPase activating protein FLCN[6,7]. TFEB lacks the TOR signalling motif responsible for the recruitment of other mTORC1 substrates. We used cryogenic-electron microscopy to determine the structure of TFEB as presented to mTORC1 for phosphorylation, which we refer to as the 'megacomplex'. Two full Rag–Ragulator complexes present each molecule of TFEB to the mTOR active site. One Rag–Ragulator complex is bound to Raptor in the canonical mode seen previously in the absence of TFEB. A second Rag–Ragulator complex (non-canonical) docks onto the first through a RagC GDP-dependent contact with the second Ragulator complex. The non-canonical Rag dimer binds the first helix of TFEB with a RagC[GDP]-dependent aspartate clamp in the cleft between the Rag G domains. In cellulo mutation of the clamp drives TFEB constitutively into the nucleus while having no effect on mTORC1 localization. The remainder of the 108-amino acid TFEB docking domain winds around Raptor and then back to RagA. The double use of RagC GDP contacts in both Rag dimers explains the strong dependence of TFEB phosphorylation on FLCN and the RagC GDP state.

TFEB is one of four members of the microphthalmia family of basic helix-loop-helix leucine zipper (bHLH-Zip) transcription factors[8]. Overexpression of TFEB promotes degradation of long-lived proteins[9], lipid droplets[10] and damaged mitochondria[11], and can induce lysosomal exocytosis[12]. Indeed, data from cellular and mouse models show that TFEB activation increases autophagic and lysosomal clearance capacity, and is therefore a potential therapeutic target for the treatment of lysosomal storage disorders[13,14] and neurodegenerative diseases involving damaged organelles and accumulation of protein aggregates[15–18]. The last include Parkinson's Disease and Alzheimer's Disease. TFEB is regulated by cellular nutrient status through the phosphorylation at several serine residues, including Ser122, Ser142 and Ser211, by the mechanistic target of rapamycin complex 1 (mTORC1) under nutrient-replete conditions[2–5]. Phosphorylation of these sites allows TFEB cytosolic retention and inactivation[3,4,19]. mTORC1 is recruited to the lysosomal membrane for activation by the Rag GTPases, which are heterodimers composed of Rag A or B bound to Rag C or D[20–22]. The pentameric Ragulator–Lamtor complex, composed of Lamtor1–5 proteins, is a scaffold that anchors the Rag GTPases to the lysosomal membrane through myristoyl and palmitoyl posttranslational modifications of its Lamtor1 subunit[23].

The cryogenic-electron microscopy (cryo-EM) structures of Rag dimers bound to mTORC1 (ref. [24]) or its Raptor subunit[25] showed that RagA[GTP] extensively contacts Raptor. Despite the importance of RagC[GDP]

in mTORC1 physiology, these structures also showed RagC[GDP] interacts with Raptor to a lesser degree and without stringent dependence on the RagC nucleotide state. The tumour suppressor FLCN is the GTPase activating protein (GAP) for RagC[26]. FLCN activity is required for the phosphorylation of TFEB, but not for other mTORC1 substrates[7]. FLCN is maintained in the inactive lysosomal FLCN complex (LFC) during amino acid starvation[6]. FLCN is reactivated under amino acid replete condition when the LFC is destabilized by the amino acid transporter SLC38A9 (ref. [27]). TFEB phosphoregulation accounts for the tumour suppressor function of FLCN in Birt–Hogg–Dubé (BHD) syndrome[7]. TFEB lacks the TOR signalling (TOS) motif found in other mTORC1 substrates[28], which enables presentation of substrates to the catalytic subunit by Raptor[29]. Instead, TFEB was shown to interact with the Rag GTPases[30], which serve as a substrate recruitment mechanism that allows TFEB phosphorylation by mTORC1 (ref. [7]). We therefore proposed that a unique structural platform directly involving RagC[GDP] might be responsible for selectively presenting TFEB as a substrate of mTORC1. We set out to test the hypothesis by reconstituting and determining the structure of the complex.

## Cryo-EM of Raptor–TFEB–Rag–Ragulator

A complex of TFEB–RagA–RagC was obtained by co-expression of full-length TFEB (R245-247A, S211A) and Rag GTPases (RagA[Q66L], RagC[S75N])

[1]Department of Molecular and Cell Biology, University of California Berkeley, Berkeley, CA, USA. [2]California Institute for Quantitative Biosciences, University of California, Berkeley, CA, USA. [3]Telethon Institute of Genetics and Medicine (TIGEM), Naples, Italy. [4]Medical Genetics Unit, Department of Medical and Translational Science, Federico II University, Naples, Italy. [5]Institute of Cell Biology, Biocenter, Medical University of Innsbruck, Innsbruck, Austria. [6]Department of Molecular and Human Genetics, Baylor College of Medicine, Houston, TX, USA. [7]Jan and Dan Duncan Neurological Research Institute, Texas Children's Hospital, Houston, TX, USA. [8]SSM School for Advanced Studies, Federico II University, Naples, Italy. [9]Helen Wills Neuroscience Institute, University of California, Berkeley, Berkeley, CA, USA. ✉e-mail: ballabio@tigem.it; jimhurley@berkeley.edu

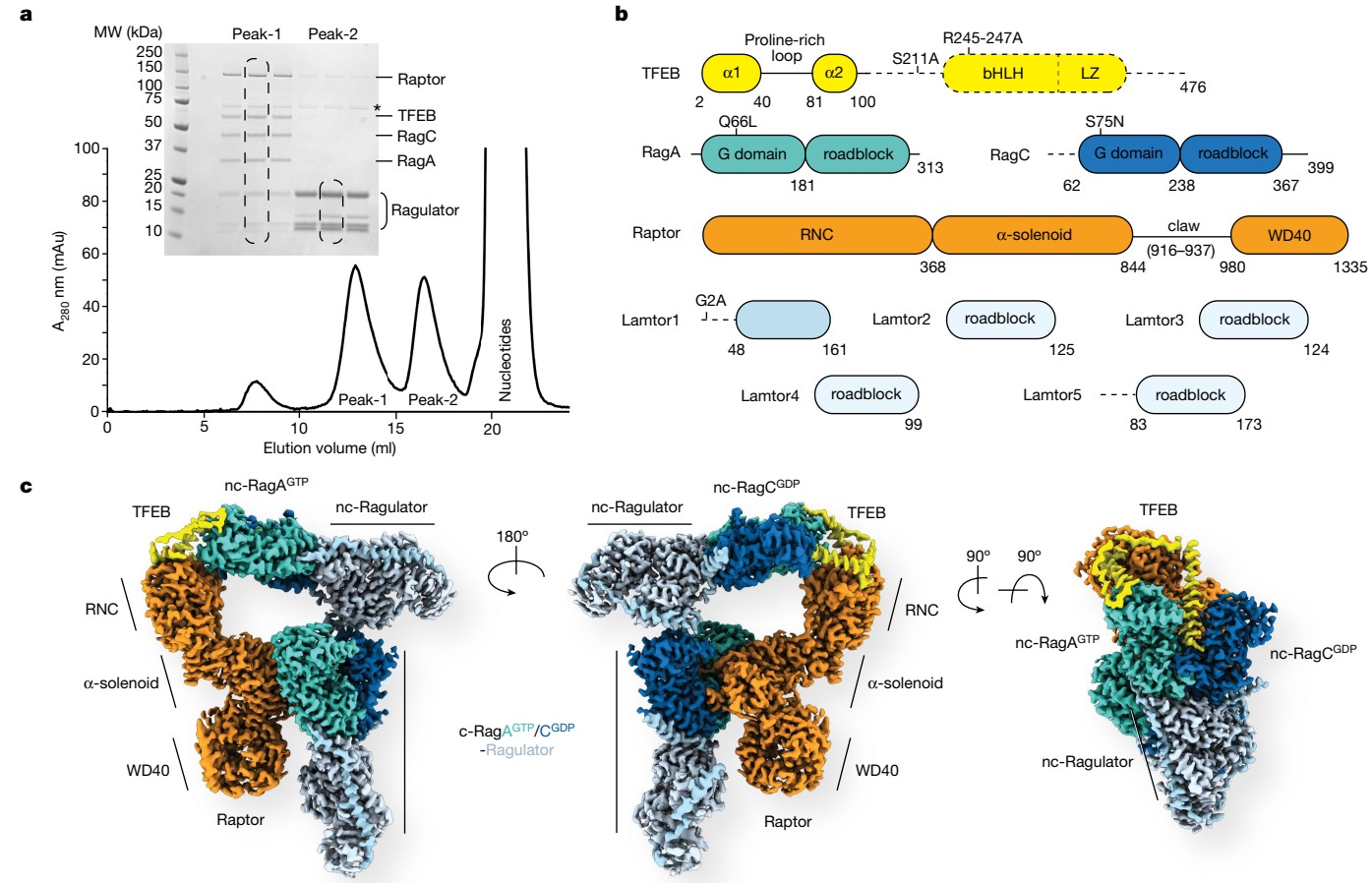

**Fig. 1 | Reconstitution and structure of the Raptor–TFEB–Rag–Ragulator complex. a**, Size-exclusion chromatography and SDS–PAGE of assembled Raptor–TFEB–Rag–Ragulator complex. Peak-1 corresponds to the fully assembled complex, and peak-2 represents Ragulator alone. All the corresponding bands are labelled, the asterisk indicates HSP70 contamination. MW, molecular weight; $A_{280}$, absorbance at 280 nm. **b**, Domain arrangement of all the subunits in the complex. Unresolved domains are indicated by dashed lines. **c**, A composite cryo-EM density map of the complex, assembled from three focused-refinement maps (Raptor, c-RagA$^{GTP}$/RagC$^{GDP}$–Ragulator and TFEB-nc–RagA$^{GTP}$/RagC$^{GDP}$– Ragulator). Different contour levels were used for optimal visualization using UCSF ChimeraX[46]. c, canonical; nc, non-canonical.

in human embryonic kidney (HEK)-293F GnTI⁻ cells, and found to be stable under size-exclusion chromatography (Extended Data Fig. 1a,b). Mutations in the nuclear localization signal of TFEB (R245A, R246A, R247A) were introduced to prevent nuclear translocation[30] during expression in HEK cells. We also introduced the S211A mutation in TFEB because it had been reported to stabilize TFEB association with the Rag GTPases in cells[30]. The mutations RagA (Q66L) and RagC (S75N) were incorporated to promote the active configuration of Rag GTPases (RagA$^{GTP}$–RagC$^{GDP}$). We reconstituted the TFEB–Rag complex with purified Ragulator complex and solved the cryo-EM structure (Extended Data Fig. 1a,c–e). However, no TFEB density was observed (Extended Data Fig. 1f,g), indicating that further interactions were required to stabilize TFEB for structural studies. Previous structures[24,25] suggested that Raptor would be structurally proximal to the TFEB binding site. We then repeated the reconstitution in the presence of purified Raptor, assembled a Raptor–TFEB–Rag–Ragulator complex (Fig. 1a), and determined its cryo-EM structure to an overall 3.1-Å resolution (Extended Data Fig. 2d and Extended Data Table 1).

The structure of Raptor–TFEB–Rag–Ragulator showed a 2:1 stoichiometry for Rag–Ragulator with respect to Raptor, as compared to 1:1 observed in the structures of mTORC1/Raptor–Rag–Ragulator[24,25] (Fig. 1c). We refer to the Rag–Ragulator module that binds to Raptor as previously reported as the 'canonical' (c) Rag–Ragulator, and the

second module as the 'non-canonical' (nc) Rag–Ragulator. Further local refinement of Raptor, canonical Rag–Ragulator and TFEB-nc–Rag–Ragulator yielded 2.8, 2.9 and 2.9 Å cryo-EM maps, respectively, which allowed us to build accurate atomic models (Extended Data Fig. 2a,d). The cryo-EM density of TFEB was clearly visible and corresponds to residues 2–105 (Extended Data Fig. 3a). The N and C termini of this segment are helical and referred to as α1 and α2, and they are connected to a Pro-rich loop. The TFEB-nc–Rag–Ragulator spans the Raptor and the canonical Rag dimer, forming a closed triangle with extensive contacts at both ends (Extended Data Fig. 4a,b). Both Rag heterodimers are in active states, on the basis of the high-resolution cryo-EM density of the corresponding nucleotide (Extended Data Fig. 3b). The conformations of the c-Rag–Ragulator and nc-Rag–Ragulator complexes are virtually identical (Extended Data Fig. 4c).

## Interactions between TFEB and Rags

The structure showed that TFEB residues 2–105 were ordered, suggesting that these residues were both necessary and sufficient to form a stable complex with active Rag GTPases, and we confirmed that a slightly longer TFEB 1–109 construct was competent to form such a complex in vitro (Extended Data Fig. 5). The first 40 residues of TFEB form a long helix (α1) and occupy the cleft between the two G domains

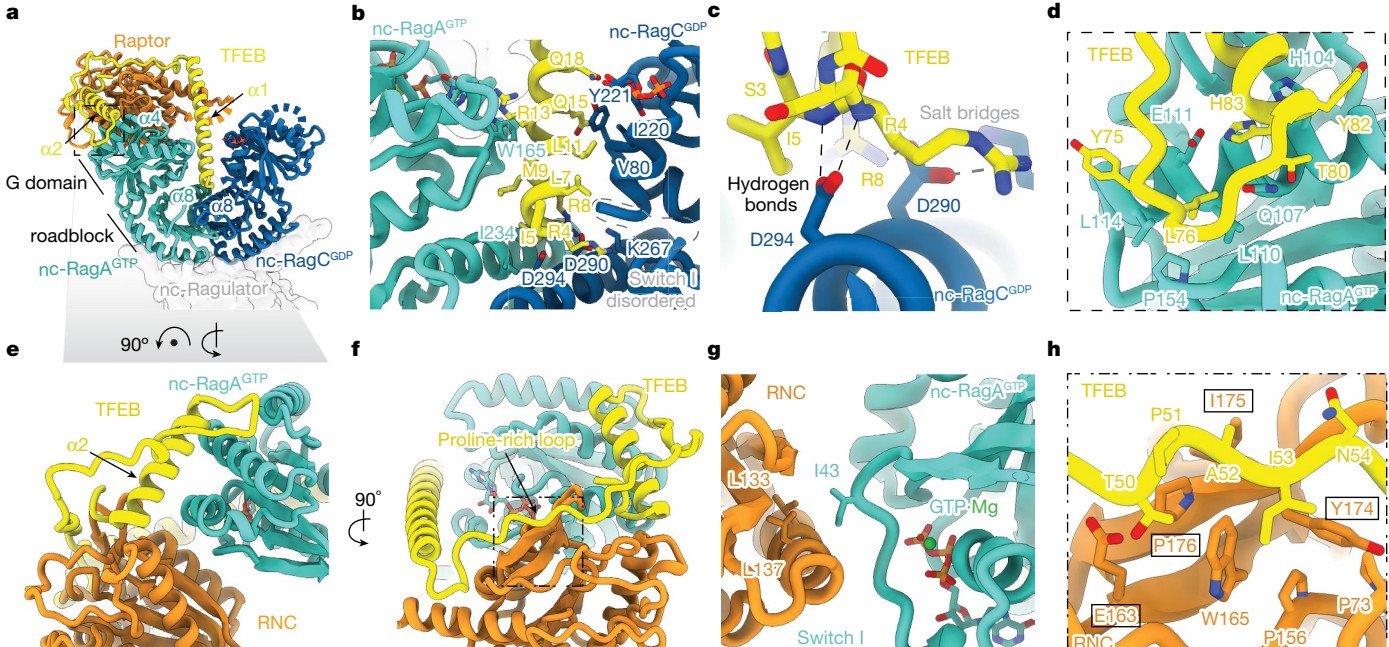

**Fig. 2 | TFEB interacts with both nc-Rag GTPases and Raptor. a**, Overall interaction between TFEB and nc-Rag GTPases is shown as ribbon models from the front view. nc-Ragulator is shown as transparent surfaces. Disordered switches I and II of nc-RagC[GDP] are shown with dashed lines. **b**, Interactions between TFEB and inter-Rag G domains at the dimer interface. **c**, Close-up view of the interaction between TFEB N terminus and α8 of RagC[GDP]. Hydrogen bonds and salt bridges are labelled and indicated with black and grey dashed lines, respectively. **d**, Close-up view of the interaction between TFEB and outer-G domain of RagA[GTP] as outlined in **a**. **e**, Ribbon model showing the interactions among TFEB, Raptor and nc-RagA[GTP]. TFEB bridges the interaction between Raptor and nc-RagA[GTP] through its Pro-rich loop and α2 region. **f**, 90°-rotated view of **e** shows the interaction between the Pro-rich loop of TFEB and RNC domain. **g**, Close-up view of the interaction between RNC domain and nc-RagA[GTP]. Ordered switch I of nc-RagA[GTP] facilitates its interaction with the RNC domain. **h**, Close-up view as outlined in **f** shows the residues responsible for the interaction between [50]TPAI[53] of TFEB and RNC domain. Raptor residues that interact with the RAIP motif of 4E-BP1 are highlighted with boxes[33].

of the non-canonical dimer of RagA[GTP] and RagC[GDP] (Fig. 2a). Residues 2–18 of TFEB α1 are embedded in the inter-Rag G domain cleft and form a roughly 570 Å interaction interface with the RagC[GDP] G domain (Fig. 2b and Extended Data Fig. 4b). The N terminus of TFEB α1 sits directly on top of the α8 of RagC[GDP] roadblock domain, which is, in turn, at the dimerization interface of Rag heterodimer. TFEB helix α1 is thus clamped within the G domain cleft by hydrogen bonds between Asp[294] of RagC[GDP] and the backbone of Arg[4] and Ile[5] of TFEB, and salt bridges between Asp[290] of RagC[GDP] and Arg[4], Arg[8] of TFEB (Fig. 2c). The α1 of TFEB also contacts RagC[GDP] through hydrophobic interactions between Leu[7], Leu[11] of TFEB and Val[80], Ile[220] of RagC[GDP] and between the side chain of Gln[15] of TFEB and Tyr[221] of RagC[GDP] (Fig. 2b). A stacking interaction is also present between the carbon chain of Arg[13] of TFEB and Trp[165] of RagA[GTP]. There are fewer contacts between TFEB α1 and RagA[GTP], primarily through hydrophobic interactions between Ile[5], Met[9] of TFEB and Ile[234] of RagA[GTP] (Fig. 2b). These data show how the TFEB N terminus is clamped between the Rag G domains in a strictly RagC[GDP]-dependent manner.

To validate the functional role of the TFEB N terminus, TFEB mutants I5D, L7D/R8D and M9D were expressed in HeLa cells as green fluorescent protein (GFP) fusions. Cytosolic localization and Ser211 phosphorylation of wild-type TFEB was observed robustly in the absence of Torin1, as expected (Fig. 3a,b). Instead, all three of the mutants showed constitutively nuclear localization and defective Ser211 phosphorylation, even in the absence of Torin1 (Fig. 3a,b). TFEB lysosomal localization and mTORC1 interaction, which are observed for wild-type TFEB in Torin1-treated cells, were essentially abolished for all the mutants (Fig. 3c,d), confirming the functional requirement for these extreme N-terminal residues of TFEB in the G domain clamp. Next, we used a HeLa RagC knockout (KO) cell line to validate the role of the RagC clamp. The interaction between TFEB and the transiently transfected

Rag GTPases was significantly impaired in both RagC[D290R] and RagC[D294R] in comparison to wild-type RagC (Fig. 3e). Moreover, expression of RagC[D294R] in the RagC KO cells prevented TFEB phosphorylation in amino acid replete cells, even in the absence of Torin1 (Fig. 3f). However, the phosphorylation of S6K and 4E-BP1 was normal, indicating that these TOS motif substrates do not require the RagC clamp. RagC[D294R] expression supported mTOR–RagC colocalization in amino acid replete conditions (Fig. 3k), but not cytosolic localization and RagC colocalization of TFEB in the absence and presence of Torin1, respectively (Fig. 3i,j). These data support that the RagC G domain clamp uniquely regulates TFEB phosphorylation.

Notably, a previously unknown interaction interface between residues 76–83 of TFEB and α4 of RagA[GTP] is also shown in our structure (Fig. 2d). Leu[76] of TFEB inserts into a hydrophobic pocket formed by Leu[110], Leu[114], Pro[154] and Leu[155] of RagA[GTP]. A salt bridge between His[84] of TFEB and Glu[111] of RagA[GTP] is also observed. We then used a RagA KO cell line to validate whether the structural analysis faithfully reflects cell physiology. The interaction between TFEB and the transiently transfected Rag GTPases was significantly impaired in the RagA triple mutant (H104D/Q107R/E111R) in comparison to wild-type RagA (Fig. 3g). The expression of RagA[H104D/Q107R/E111R] prevented TFEB phosphorylation in amino acid replete cells (Fig. 3h), but did not affect the phosphorylation of S6K and 4E-BP1. In addition, RagA[H104D/Q107R/E111R] expression supported mTOR–RagA colocalization in amino acid replete conditions (Fig. 3n), but not cytosolic localization and RagA colocalization of TFEB in the absence and presence of Torin1, respectively (Fig. 3l,m). Therefore, the unique RagA–TFEB interaction is necessary for TFEB phosphorylation.

TFEB interacts only with the active RagC[GDP]-containing Rag dimer. Alignment of the TFEB bound active Rag GTPases and inactive Rag GTPases (RagA[GDP]–RagC[GTP])[6,27,31,32] based on the α8 of RagC roadblock domains indicated that the switch I of RagC[GTP] sterically clashes with the

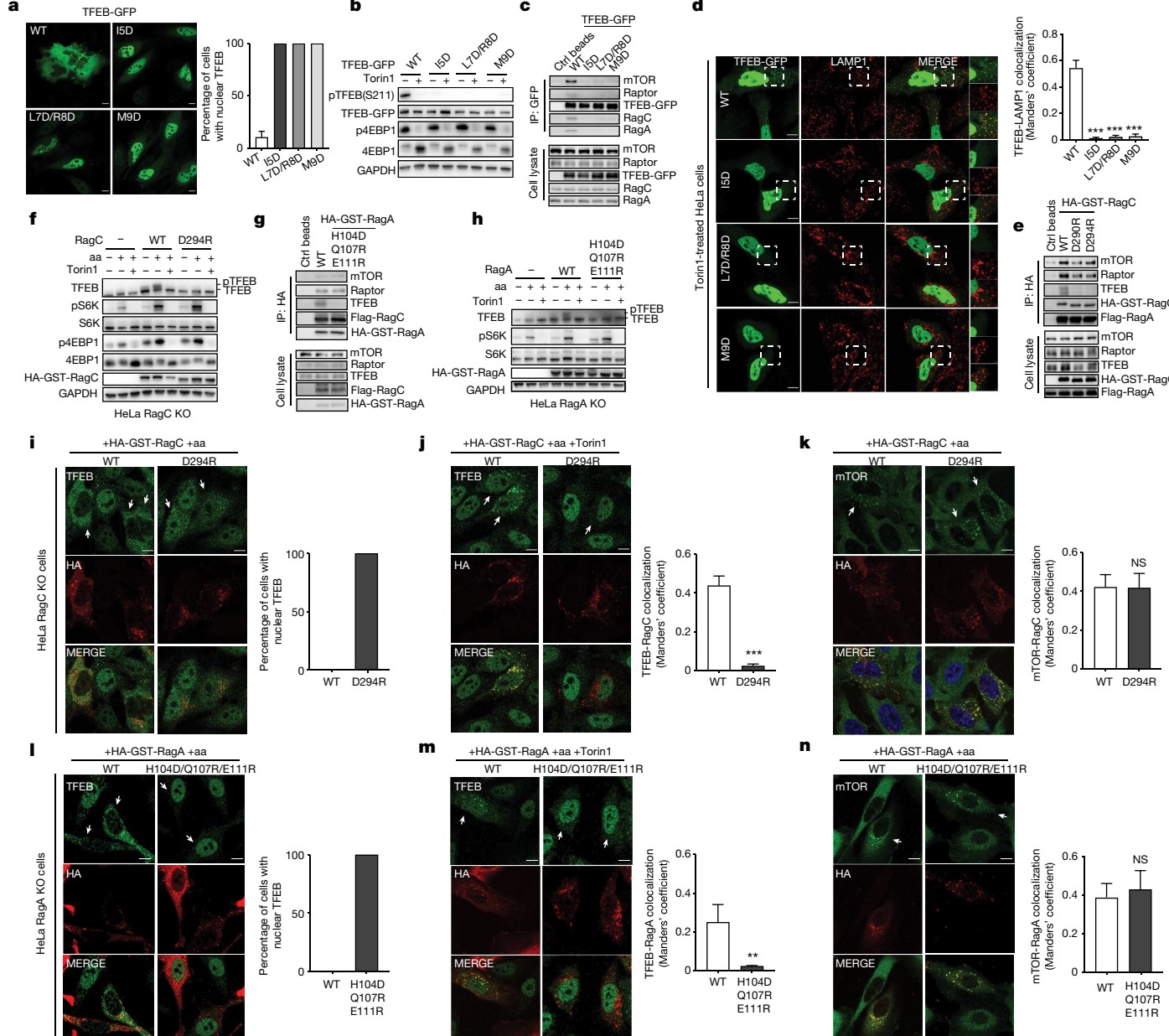

**Fig. 3 | Function of the TFEB-nc-Rag GTPases interface. a**, Cells expressing wild-type (WT) or mutant TFEB-GFP were analysed using immunofluorescence to determine the percentage of cells showing nuclear TFEB, shown as mean ± s.e. throughout; $n = 12$ independent fields per condition. **b**, Immunoblot of HeLa cells expressing wild-type or mutant TFEB-GFP. **c**, Representative co-immunoprecipitation of Flp-In 293 T-REx cells transfected with wild-type or mutant TFEB-GFP. **d**, Microscopy analysis of Torin1-treated HeLa cells, $n ≥ 5$ independent fields per condition. ***$P ≤ 0.0001$ throughout. One-way analysis of variance (ANOVA), Dunnett's multiple comparisons test. **e**, Representative co-immunoprecipitation of HeLa RagC KO cells transfected with the indicated constructs. **f**, Immunoblot of RagC KO HeLa cells transfected with empty vector or wild-type RagC or RagC(D294R). Cells were amino acid starved and refed in the presence or absence of 250 nM Torin1. **g**, Representative co-immunoprecipitation of HeLa RagA KO cells transfected with the indicated constructs. **h**, Immunoblot of RagA KO HeLa cells transfected with empty vector or wild-type RagA or RagA(H104D/Q107R/E111R). **i**–**k**, Cells as in **f** were analysed using immunofluorescence and the percentage of the cells were determined to show nuclear TFEB (**i**) ($n = 5$ fields per condition); TFEB–RagC colocalization (**j**) ($n ≥ 5$ fields per condition, ***$P ≤ 0.0001$, unpaired $t$-test) and mTOR–RagC colocalization (**k**) ($n ≥ 12$ fields per condition, unpaired $t$-test). **l**–**n**, Cells as in **h** were analysed using immunofluorescence and quantified to calculate the percentage of the cells showing nuclear TFEB (**l**) ($n ≥ 4$ independent fields per condition), TFEB–RagA colocalization (**m**) ($n ≥ 4$ fields per condition, **$P ≤ 0.002$, unpaired $t$-test) and mTOR–RagA colocalization (**n**) ($n = 5$ independent fields per condition, unpaired $t$-test). Scale bar, 10 μm. NS, not significant; aa, amino acid; Ctrl, control; HA, haemagglutinin; IP, immunoprecipitation; GST, glutathione $S$-transferase.

α1 of TFEB, therefore precluding TFEB binding (Extended Data Fig. 6a). In addition, the wide opening conformation of G domains in the inactive Rag GTPases (RagA^GDP–RagC^GTP) effectively prevent the interaction between TFEB and the Rag heterodimer in the inactive state. Alignment of the TFEB bound active RagA^GTP and inactive RagA^GDP based on the

α4 of RagA G domain showed apparently trivial structural differences in the TFEB binding region (Extended Data Fig. 6b), suggesting the nucleotide loading state of RagA does not impose selectivity towards TFEB binding at the unique RagA–TFEB interface. However, RagA^GTP is required for mTORC1 recruitment to the lysosome and subsequent

phosphorylation of both TFEB and TOS motif-containing substrates. In summary, RagC[GDP] and the unique RagA–TFEB interaction are specifically important for TFEB phosphorylation.

## Bridging Rag–Ragulator complexes

The TFEB-nc–Rag–Ragulator complex is stabilized by interactions at both ends. On one end, the Pro-loop and α2 of TFEB bridge the Raptor N-terminal conserved (RNC) domain and the ordered switch I of nc-RagA[GTP] (Fig. 2e,f). As the switch I of RagA is disordered in the GDP-bound state[6,27], it emphasizes the importance of nc-Rag GTPases being in the active state (RagA[GTP]–RagC[GDP]). The contact between Raptor and nc-RagA[GTP] is maintained by hydrophobic interaction between Leu[133], Leu[137] of Raptor and Ile[43] of nc-RagA[GTP] (Fig. 2g). The residues Thr[50], Pro[51], Ala[52] and Ile[53] of TFEB (TPAI) cover a patch on the RNC domain, formed by residues Pro[73], Pro[156], Glu[163], Trp[165], Tyr[174], Ile[175] and Pro[176] (Fig. 2h). The mTORC1 substrate 4E-BP1 contains an N-terminal RAIP sequence that binds to the same site on the RNC domain[33]. TFEB Ala[52] and Ile[53] also interact with Raptor residues Tyr[174] and Ile[175] by β strand augmentation, as had been proposed for the 4E-BP1 RAIP–RNC interaction[33]. However, mutational disruption of the [50]TPAI[53] sequence had no effect on TFEB phosphorylation or subcellular localization (Extended Data Fig. 7d,e).

On the other side of the interface, the Lamtor1 subunit of the non-canonical Ragulator makes close contacts with the G domain of canonical RagC[GDP] (Fig. 4a). The α2 of Lamtor1 resides on top of the α4 and α5 helices of RagC, stabilized by Met[82] of Lamtor1 inserting into the hydrophobic pocket formed by Tyr[150], Met[151], Leu[154] of RagC[GDP] (Fig. 4b). Salt bridges are formed between Asp[83], Arg[86] of Lamtor1 and Arg[198], Asp[202] of RagC[GDP], respectively. These data indicate that the RagC[GDP] state in the canonical Rag dimer is important for nc-Lamtor1 interaction and stabilization of the nc-Rag–Ragulator binding, thus heightening the sensitivity of the entire assembly to the RagC nucleotide state and thus the dependency of TFEB phosphorylation on FLCN.

To validate the role of the c-RagC and nc-Ragulator bridging interaction, RagC residues Tyr[150], Met[151] and Arg[198] were mutated and expressed in RagC KO HeLa cells. The constructs RagC[Y150D], RagC[Y150D/R198D] and RagC[Y150D/M151D/R198D] all showed normal S6K and 4E-BP1 phosphorylation in amino acid replete conditions, but reduced or no TFEB phosphorylation under these same conditions (Fig. 4c). Consistently, only wild-type RagC restored TFEB cytosolic localization in RagC KO cells (Fig. 4e). Furthermore, in contrast to wild-type RagC, none of the RagC mutants were able to restore TFEB–RagC colocalization in Torin1-treated RagC KO cells (Fig. 4f). Wild-type and all three RagC mutants were all, however, able to restore mTOR lysosomal localization in RagC KO cells (Fig. 4g), consistent with a selective role for this interaction in regulating the phosphorylation of TFEB, but not other mTORC1 substrates. Transfected RagC[Y150D] and RagC[Y150D/R198D] had no defect in their interactions with TFEB, Raptor or mTOR, but a modest defect was noted for RagC[Y150D/M151D/R198D] (Fig. 4d). The co-immunoprecipitation results confirm that the megacomplex can still be formed in bulk, whereas the microscopy results indicate that significant recruitment of TFEB at any given time point is low because of decreased affinity for Ragulator, which in turn effectively prevents TFEB phosphorylation.

## Cryo-EM structure of the megacomplex

To understand how TFEB is phosphorylated by mTORC1 as presented by the Raptor–Rag–Ragulator complex, we reconstituted mTORC1–TFEB–Rag–Ragulator megacomplex and determined its structure by cryo-EM. Two main populations of the megacomplex were resolved, showing that either one or two copies of TFEB are present on mTORC1 (Extended Data Fig. 8a). The reconstruction in C2 symmetry for the mTORC1 with two copies of TFEB resulted in a 3.7-Å resolution cryo-EM map. Further symmetry expansion and local refinement of the asymmetric unit

improved the resolution to 3.2 Å (Extended Data Fig. 8e). A composite map was generated by superimposing the local reconstructions to the C2 symmetric reconstruction (Fig. 5a). We then built the atomic model for the entire complex, which contains mTOR, Raptor, mLST8, TFEB, active Rag GTPases, Ragulator with a stoichiometry of 2:2:2:2:4:4, containing a total of 36 polypeptide chains (Fig. 5b, Extended Data Table 1 and Supplementary Video 1).

The long axis of the megacomplex is about 370 Å, and shows a curved geometry, with the HEAT domain and FAT domain of mTOR facing the convex and concave side, respectively. The binding mode of TFEB to the Rag GTPases and Raptor is essentially unchanged in the presence of the entire mTORC1 complex. Residues 2–108 of TFEB were resolved in the cryo-EM structure, essentially as before, but with the rest of TFEB not visualized and an empty active site despite the presence of a non-hydrolysable ATP analogue and the presence of sequences containing phosphorylation sites (Extended Data Fig. 9a,b). The inability to visualize these regions is probably due to inherent flexibility in the sequences containing phosphorylation sites. The Pro-loop and α2 of TFEB are positioned near the active site of mTOR, surrounded by the FKBP12-rapamycin-binding (FRB) domain, the kinase domain (KD) N lobe and C lobe (Fig. 5c). TFEB shows limited overlap with the PRAS40 (ref. [29]), but no overlap with the S6K, at the FRB binding site (Extended Data Fig. 9c,d). In addition, Tyr[95] and Thr[99] of TFEB are close to, but do not directly contact, a hinge loop (residues 2115–2118) at the end of mTOR FRB domain (Fig. 5d)

Whereas the last ordered residue of TFEB is roughly 34 Å from the catalytic residue Asp[2338] of mTOR (Fig. 5d), this still places Ser122 and Ser142 of TFEB close enough for delivery to the active site. In principle, the predicted flexible sequence from 109–210 could be long enough to deliver Ser211 to the second (distal) active site in the mTOR dimer, which is roughly 116 Å (linear distance) from TFEB residue 108. TFEB phosphorylation by mTORC1 is independent of Rheb[7]. In the case of Rheb-dependent mTORC1 activation, a large-scale conformational change of mTORC1 was observed in the Rheb-bound state, which is thought to be essential for phosphorylation of TOS-containing substrates[29]. Our results shed light on the alternative structural mechanism of Rheb-independent TFEB phosphorylation by mTORC1, with compatible anchoring geometry on the lysosomal membrane (Extended Data Fig. 10).

## Discussion

The work described above provides structural evidence in support of the findings that TFEB phosphorylation by mTORC1 is uniquely regulated through an elaborate RagC[GDP]-dependent mechanism[7]. The RagC[GDP] state is required, in the first instance, to maintain favourable contacts with the TFEB N-terminal helix in the non-canonical Rag dimer. In the canonical Rag dimer, the RagC[GDP] state is also important, in this case, for recruitment of the non-canonical Ragulator to the rest of the complex. The elaborate complex involved in presenting TFEB for phosphorylation by mTORC1 thus depends on two molecules of RagC[GDP]. This mechanism serves to increase the stringency of TFEB regulation by the RagC GAP, FLCN. In the past few years, it has emerged that FLCN controls the phosphorylation of TFEB and other microphthalmia-associated transcription factors, but not that of many other well-known mTORC1 substrates such as S6 kinase, ULK1 and 4E-BP1 (refs. [6,7,34,35]). As the tumour suppressor whose mutation is responsible for BHD syndrome[36], FLCN is under exceptionally tight regulation. An elaborate set of structural gymnastics keep FLCN inactive by retaining it in the lysosomal LFC under starvation conditions[6], and reactivating it through intercleft competition with nutrient-activated SLC38A9 on refeeding[27]. The intercleft binding site for TFEB in the nc-Rag dimer overlaps with the site occupied by FLCN-FNIP2 in the LFC[6,31] and with SLC38A9 (ref. [27]), highlighting the complex time-sharing of different regulatory factors in the cleft. To this picture of stringent

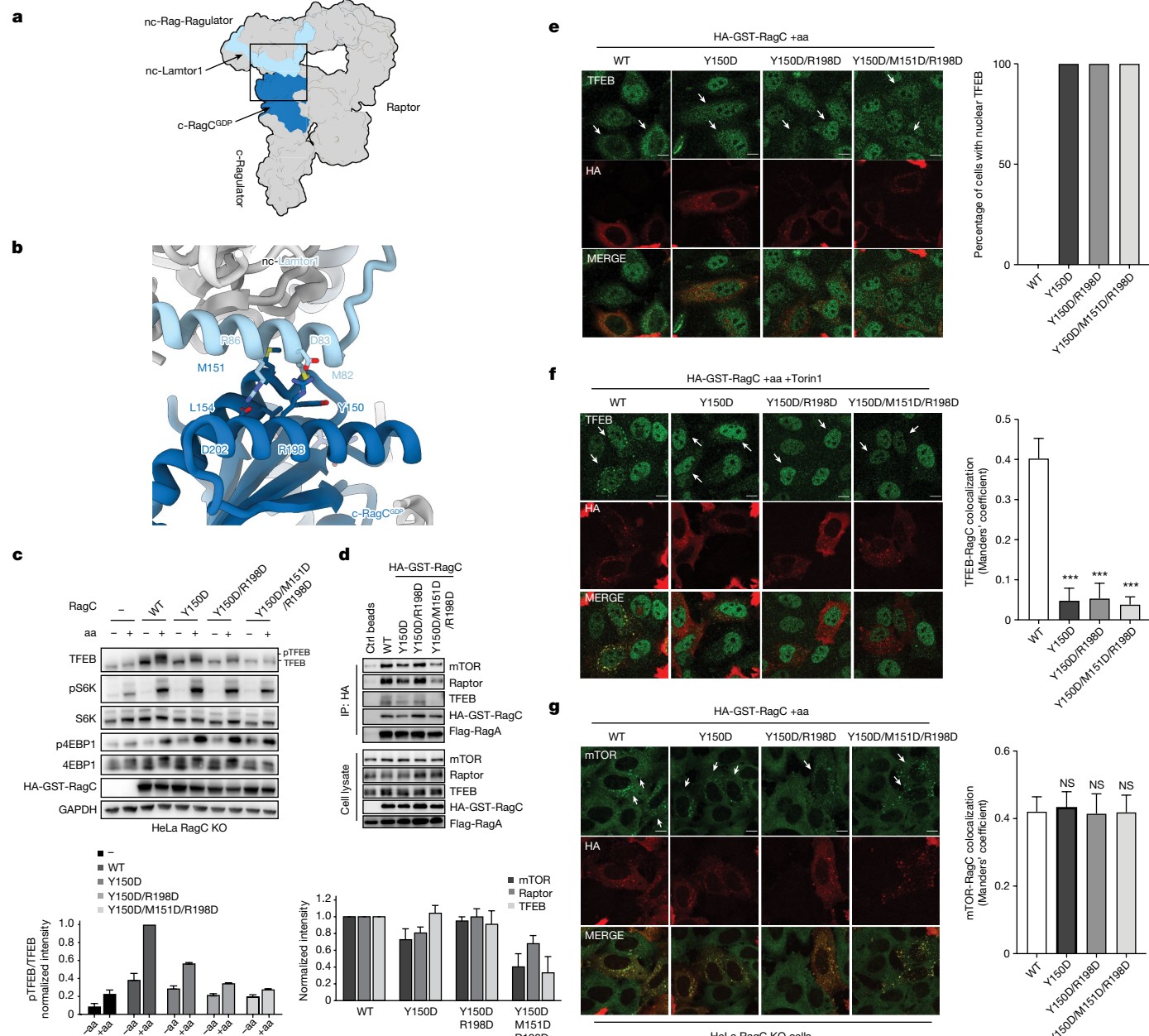

**Fig. 4 | Function of the nc-Ragulator and c-RagC^GDP interface. a**, Cartoon representation that highlights the interacting subunits at the end with nc-Ragulator. **b**, Close-up view as outlined in **a** shows the residues responsible for the interaction between nc-Lamtor1 and c-RagC^GDP. **c**, Representative immunoblot of RagC KO HeLa cells transfected with empty vector or wild-type RagC or RagC mutants (Y150, Y150/R198D or Y150/M151/R198D). Cells were amino acid starved and refed in the presence or absence of 250 nM Torin1. Quantifications are shown with mean ± s.e. throughout; *n* = 2 experiments.

**d**, Representative co-immunoprecipitation of HeLa RagC KO cells transfected with the indicated constructs; *n* = 3 experiments. **e**–**g**, Cells as in **c** were analysed using immunofluorescence and quantified to calculate the percentage of the cells showing nuclear TFEB (**e**) (*n* = 5 independent fields per condition); TFEB–RagC colocalization (**f**) (*n* ≥ 5 independent fields per condition, ***P ≤ 0.0001, one-way ANOVA, Dunnett's multiple comparisons test) and mTOR–RagC colocalization (**g**) (*n* ≥ 8 independent fields per condition; NS, not significant; one-way ANOVA, Dunnett's multiple comparisons test). Scale bar, 10 μm.

regulation of FLCN, we have now added an even more elaborate mode of regulation of TFEB phosphorylation downstream of FLCN.

The structures described here confirm several predictions in the literature while also showing many unexpected features. The interaction of Raptor with the canonical Rag–Ragulator complex was just as expected[24,25]. The N-terminal 30 amino acids of TFEB were correctly predicted to be essential for its Rag binding, lysosomal localization and mTORC1 phosphorylation[30]. The role of the subsequent 80 TFEB residues, however, was unanticipated. The presence of direct TFEB–Raptor interactions was completely unexpected. An Ile- and

Pro-containing TPAI motif of TFEB binds directly to a surface patch on the RNC domain of Raptor that was previously shown to bind the RAIP sequence of 4E-BP1 (ref. [33]). The TOS-containing mTORC1 inhibitor PRAS40 also contains an RAIP-like motif that interacts with the same patch on the RNC domain[24], although its functional role has not been assessed. We found that a quadruple TPAI mutant of TFEB has no effect on TFEB phosphorylation by mTORC1 and subcellular localization. This is consistent with the finding that TFEB is not stably bound to Raptor in RagA/B deficient cells[7], in the presence of inactive Rag GTPases[2,5], or when the Rag-binding N-terminal residues of TFEB are absent[3,7,37].

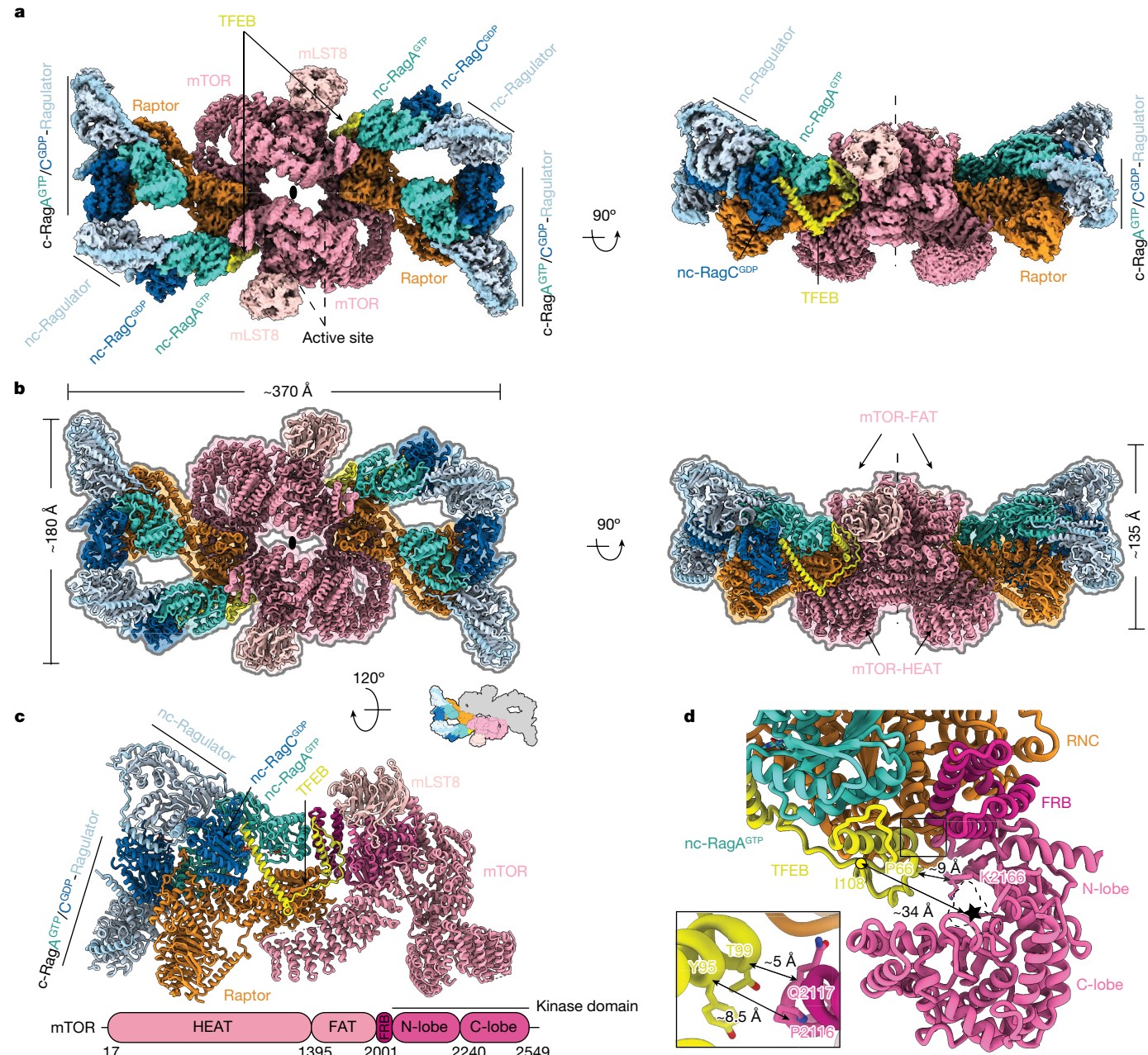

**Fig. 5 | Structure of the mTORC1–TFEB–Rag–Ragulator megacomplex.**
**a**, Composite cryo-EM density map of the dimeric mTORC1–TFEB–Rag–
Ragulator megacomplex shown from top and side views. The active sites of
mTOR are labelled with dashed arrows. The twofold axis is labelled as an oval
symbol in the top view and a dash line in the side view. Different contour levels
were used for optimal visualization using UCSF ChimeraX[46]. **b**, Atomic model of
the dimeric megacomplex shown in the same orientation as in **a. c**, The ribbon
model of an asymmetric unit. The domain organization of mTOR is shown.

**d**, Focused view of the active site of mTOR, the HEAT and FAT domains are
omitted for clarity. The ATP binding site is outlined with a dashed line.
The distance between Pro[66] of TFEB and Lys[2166] of mTOR is drawn with a
double-headed arrow. The distance between Ile[108] of TFEB and the active site
of mTOR is calculated on the basis of the distance between Ile[108] of TFEB and
Asp[2338] of mTOR. The inset highlights the distance between TFEB and the hinge
loop (residues 2115–2118) at the end of mTOR FRB domain. Distances are
calculated on the basis of the Cα atoms.

The 4E-BP1 RAIP motif is separated from the TOS motif by 100 residues,
and the RAIP–Raptor interaction makes a secondary contribution to
mTORC1 binding relative to the TOS motif. Despite the fact structures
show that the Raptor–TFEB binding interface is a prominent feature,
the TPAI mutational data and previously published evidence indicate
that TFEB–Raptor interaction is secondary to the contacts with the Rags
and may have an auxiliary role in complex stabilization. Our structure
illuminates a direct interaction between a Rag G domain and Ragulator

(canonical RagC with non-canonical Ragulator), which we found to be
important for TFEB phosphorylation. The question of how the mega-
complex dissociates in cells following Ser211 phosphorylation remains
open. Dissociation might be mediated by phosphorylation-triggered
structural rearrangements within TFEB or other megacomplex com-
ponents, or binding to 14-3-3 proteins[2–5] or other factors.

All of the TFEB-contacting residues of RagC are conserved in RagD,
consistent with the finding that expression of either RagC or RagD can

rescue TFEB recruitment to lysosomes in RagC/D double KO cells[7]. However, a recent publication[38] suggested that RagC plays only a minor role in the recruitment and phosphorylation of TFEB. By contrast with this conclusion, here we showed that TFEB forms a stable complex with a RagA–RagC dimer, and that TFEB phosphorylation and subcellular localization are drastically affected in RagC KO cells, despite the presence of endogenous RagD. These results are in line with previous publications showing that RagC depletion promotes TFEB/TFE3 dephosphorylation and nuclear translocation[39,40] and that the expression of a constitutively active RagC mutant is sufficient to rescue TFEB/TFE3 phosphorylation and subcellular localization in FLCN KO cells[7,40]. Together, these data clearly demonstrate that RagC plays a major role in TFEB phosphorylation.

Overall, this work provides structural evidence of a non-canonical mTORC1 signalling that allows selective control of TFEB activity under specific conditions[7,39,41–44]. The hypophosphorylation and consequent hyperactivation of TFEB in the absence of FLCN drives increased mTORC1 activity and tumorigenesis in BHD syndrome[7]. Thus, in BHD, there might be therapeutic benefits to enhancing TFEB phosphorylation by bypassing RagC^GDP. The complexity and stringency of the structural mechanism for RagC^GDP-dependent phosphorylation suggests this will be challenging. On the other hand, enhanced activation of TFEB may be desirable in treating lysosomal storage disorders[13,14], promoting clearance of toxic aggregates and debris in neurons[15–18] and preventing non-alcoholic fatty liver disease through lipid clearance[45]. The structure presented here identifies several new interfaces that could be targeted to such an end.

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

## Methods

### Protein expression and purification

The full-length codon-optimized human TFEB with S211A and R245-247A mutations, human RagC with S75N mutation and human RagA with Q66L mutation were synthesized (Twist Bioscience) and cloned into a pCAG vector individually. The TFEB (S211A, R245-247A) construct included a TEV-cleavable GFP-His$_{10}$ tag at the C terminus. The RagC (S75N) construct included a tobacco etch virus (TEV)-cleavable GST tag at the N terminus, whereas the RagA (Q66L) was tagless. For the expression and purification of the TFEB–Rag GTPases complex, the HEK293F GnTI⁻ cells were transfected with a total of 1 mg of plasmid DNA (333 µg TFEB, 400 µg RagA and 267 µg RagC) and 4 mg polyethylenimine (Sigma-Aldrich) per litre at a density of $1.5–1.8 \times 10^6$ cells per ml. Cells were collected after 48 h, and lysed by gentle nutating in wash buffer (50 mM HEPES, 150 mM NaCl, 2.5 mM MgCl$_2$, 1 mM TCEP, pH 7.4) supplemented with 0.4% CHAPS and Protease Inhibitor (Roche) for 1 h. Lysate was cleared by centrifugation at 35,000$g$ for 35 min. Supernatant was incubated with glutathione Sepharose 4B (GE Healthcare) resin for 2 h. The resin was then first washed in the modified wash buffer with 200 mM NaCl and 0.3% CHAPS, and then in the wash buffer. The complex was eluted from the resin by a wash buffer with 10 mM reduced glutathione, and then incubated with TEV protease overnight. Eluted complexes were concentrated and further purified by size-exclusion chromatography using a Superose 6 10/300 GL (GE Healthcare) column equilibrated in the wash buffer. All purification steps were performed at 4 °C. Proteins were flash frozen in liquid nitrogen and stored in −80 °C. Attempted co-expression and purification of wild-type TFEB-GFP with active Rag GTPases was carried out as described above, however, wild-type TFEB-GFP did not co-elute with the Rags (Supplementary Fig. 2).

The human Ragulator complex (GST-TEV–Lamtor1, His$_6$-TEV–Lamtor2) was expressed in *Spodoptera frugiperda* (Sf9) cells through baculovirus infection and purified as previously described[47]. In brief, Sf9 cells were pelleted after 72 h of baculovirus infection and lysed in the wash buffer with 1% Triton X-100 and protease inhibitor. The cleared supernatant after centrifugation was applied to Ni-nitrilotriacetic acid gravity column (Thermo Scientific), washed with the wash buffer containing 200 mM NaCl, and eluted with the wash buffer with 250 mM imidazole. The elution was then applied to glutathione Sepharose 4B (GE Healthcare) gravity column, washed with the wash buffer. The complex was then eluted by on-column TEV cleavage overnight without nutation. Further purification was done by size-exclusion chromatography with a Superdex 200 10/300 GL column (GE Healthcare) column.

Three subunits of the human mTORC1 complex (mTOR, Raptor, mLST8) were codon-optimized and synthesized (GenScript). The mTOR gene was cloned into a pCAG vector without a tag, the Raptor gene was cloned into a pCAG vector with an uncleavable tandem 2× Strep II-1× FLAG-tag at the N terminus, and the mLST8 gene was also cloned into a pCAG vector with an uncleavable tandem 2× Strep II-1× FLAG-tag at the N terminus. The mTORC1 complex was produced in a similar manner to the TFEB–Rag GTPases complex, except that the total amount of DNA was increased to 1.35 mg (900 µg of mTOR, 250 µg of Raptor and 200 µg of mLST8) per litre of cells. The purification procedure of the mTORC1 complex is similar to that previously described[24]. On the other hand, Strep-Tactin resin (IBA Lifesciences) was used for the affinity purification and the complex was eluted with the wash buffer (50 mM HEPES, 150 mM NaCl, 1 mM TCEP, pH 7.4) containing 10 mM D-desthiobiotin. The elution was diluted into equal volume of salt-free buffer (50 mM HEPES, 1 mM TCEP, pH 7.4) and applied to a 5 ml HiTrap Q column (GE Healthcare). The mTORC1 complex and free RAPTOR were separated by a 100 ml salt gradient with salt-free buffer and high salt buffer (50 mM HEPES, 1 M NaCl, 1 mM TCEP, pH 7.4). The fractions containing mTORC1 complex and free Raptor were concentrated to 1.3 and 0.5 mg ml⁻¹, respectively. Purified proteins were flash frozen in liquid nitrogen and stored at −80 °C.

### Cryo-EM sample preparation and imaging

The Raptor–TFEB–Rag–Ragulator complex was prepared by incubating 0.48 µM Raptor, 0.59 µM TFEB–Rag GTPases, 1.42 µM Ragulator, 9.5 µM GTP and 9.5 µM GDP in the wash buffer on ice for 5 h. Further purification of the complex was achieved by running a Superose 6 10/300 GL column. Fractions containing the fully assembled Raptor–TFEB–Rag–Ragulator complex were concentrated to 0.8 mg ml⁻¹ for cryo-EM sample preparation.

The mTORC1–TFEB–Rag–Ragulator megacomplex was reconstituted in two steps. First, the TFEB–Rag–Ragulator complex was formed by incubating 2.5 µM TFEB–Rag GTPases, 7.4 µM Ragulator, 25 µM GTP and 25 µM GDP in the wash buffer on ice for 1 h. It is further purified through a Superose 6 10/300 GL column and concentrated to 1.2 mg ml⁻¹. And then, 0.36 µM mTORC1 complex, 1.8 µM TFEB–Rag–Ragulator complex, 18 µM GTP, 18 µM GDP and 36 µM AMPPNP were incubated in 100 µl of wash buffer containing 5 mM TCEP on ice for 5 h. Assembled mTORC1–TFEB–Rag–Ragulator megacomplex was further concentrated to roughly 1 mg ml⁻¹ for cryo-EM sample preparation.

Cryo-EM specimens were prepared by applying 3 µl of freshly reconstituted complex to a glow-discharged (PELCO easiGlow, 45 s in air at 15 mA and 0.37 mbar) holey carbon grid (C-flat, 2/1-3C-T) and vitrified using a FEI Vitrobot Mark IV (Thermo Fisher Scientific) after blotting for 3 s with blot force 18, two Whatman 595 papers on the sample side and one Whatman 595 paper on the back side at 6 °C with 100% relative humidity.

Cryo-EM images of the Raptor–TFEB–Rag–Ragulator complex and mTORC1–TFEB–Rag–Ragulator megacomplex were recorded under a Titan Krios G3 microscope (Thermo Fisher Scientific) equipped with a Gatan Quantum energy filter (slit width 20 eV) and operated at 300 kV. Automated data acquisition was achieved using SerialEM[48] on a K3 Summit direct detection camera (Gatan) in the super-resolution correlated-double sampling mode with a pixel size of 0.525 Å and a defocused range of −0.8 to −2.2 µm. Beam shift was enabled to encompass four exposures per hole and nine holes per stage shift. The beam intensity was adjusted to a dose rate of roughly 1 e⁻ per Å$^2$ per frame for a 50-frame video stack with a total exposure time of 7.6 s. A total of 10,080 and 17,028 video stacks were recorded for the Raptor–TFEB–Rag–Ragulator complex and the mTORC1–TFEB–Rag–Ragulator megacomplex, respectively.

Cryo-EM images of the TFEB–Rag–Ragulator complex were recorded under a Talos Arctica microscope (Thermo Fisher Scientific) operated at 200 kV. Automated data acquisition was achieved using SerialEM[48] on a K3 Summit direct detection camera (Gatan) in the super-resolution correlated-double sampling mode with a pixel size of 0.5575 Å and a defocused range of −0.8 to −2.2 µm. Beam shift was enabled to encompass nine exposures per stage shift. The beam intensity was adjusted to a dose rate of roughly 1 e⁻ per Å$^2$ per frame for a 50-frames video stack with a total exposure time of 8.6 s. A total of 3,438 video stacks were recorded for the TFEB–Rag–Ragulator complex.

### Cryo-EM data processing

Super-resolution video stacks were motion-corrected and binned 2× by Fourier cropping using MotionCor2 (ref. [49]). Motion-corrected micrographs were primarily processed following the workflow in cryoSPARC v.3 (ref. [50]).

The data processing scheme for the Raptor–TFEB–Rag–Ragulator complex and mTORC1–TFEB–Rag–Ragulator megacomplex is shown in Extended Data Figs. 2 and 8, respectively. Owing to the size of the datasets, micrographs were split and processed following the same protocol and then combined for homogeneous refinement. Contrast transfer function determination was done using patch CTF in cryoSPARC v.3. Blob picker and template picker were both used to maximize the number of initially picked particles. Two-dimensional (2D) classification was only used to remove obvious 'junk' particles (for example, ice and

chaperonin contaminants). Heterogeneous refinement following the ab initio reconstruction was used to select good particles, preserving potential particles with rare views that could not be identified in 2D classification. After extensive cleaning using 2D classification and heterogeneous refinement, particles were merged and the duplicates were removed with a 100-Å cut-off distance. Homogeneous refinement was then performed for the full dataset. Further cleaning of the full dataset was accomplished either by three-dimensional (3D) classification with the 'skip_align' option using RELION3 (ref. [51]) or 3D classification function in cryoSPARC v.3. The conversion of data files between cryoSPARC v.3 and RELION3 was done using University of California San Francisco (UCSF) pyem[52]. Local refinement was used to produce final cryo-EM maps for model building. For the mTORC1–TFEB–Rag–Ragulator megacomplex, symmetry expansion followed by local refinement was used to generate the cryo-EM map of an asymmetric unit.

In summary, a 3.6 Å resolution map was obtained from 169,720 particles for the TFEB–Rag–Ragulator complex. Three local refinement maps were resolved for the Raptor–TFEB–Rag–Ragulator complex, including Raptor (377,569 particles), canonical Rag–Ragulator (377,569 particles) and non-canonical Rag–Ragulator (273,453 particles) to the resolutions of 2.8, 2.9 and 2.9 Å, respectively. For the mTORC1–TFEB–Rag–Ragulator megacomplex, two main populations containing either one (103,274 particles) or two copies (96,166 particles) of the TFEB and non-canonical Rag–Ragulator were both resolved with C1 symmetry to the resolution of 3.8 Å. Symmetry expansion and local refinement of the population with two copies of the TFEB and non-canonical Rag–Ragulator yielded a 3.2 Å resolution map.

The overall resolution of all these reconstructed maps was assessed using the gold-standard criterion of Fourier shell correlation[53] at 0.143 cut-off[54]. Local resolution estimation[55] and local filtering were done in cryoSPARC v.3.

### Atomic model building and refinement

To build the atomic model for Raptor–TFEB–Rag–Ragulator complex, we first fit the previous Raptor–Rag–Ragulator (Protein Data Bank (PDB) 6U62) structure in our cryo-EM map as rigid body using UCSF ChimeraX[46]. The fragments of the TFEB model were initially obtained from AlfaFold2 prediction[56], and manually docked into our cryo-EM map. A composite map combining the three focused-refinement maps was assembled using PHENIX[57]. Model refinement against the composite map was performed by real-space refinement in PHENIX[58]. Manual model building was done with COOT[59] and ISOLDE[60] to inspect and improve local fitting. The iterative process of refinement and the manual building was conducted to achieve the best model. For the mTORC1–TFEB–Rag–Ragulator megacomplex, the refined Raptor–TFEB–Rag–Ragulator and previous mTORC1 (PDB 6BCX) structures were docked in our cryo-EM map. A composite map of the symmetric complex using the focused-refinement asymmetric unit was generated. The same model building procedure was performed as described above. All the figures and videos were made using UCSF ChimeraX.

### Materials and plasmids for cellular assays

Reagents used in this study were obtained from the following sources: antibodies to mTOR (catalogue no. 2983, 1:100 immunofluorescence), Phospho-p70 S6 Kinase (Thr389) (1A5) (catalogue no. 9206, 1:1,000 western blot), p70 S6 Kinase (catalogue no. 9202, 1:1,000 western blot), 4E-BP1 (catalogue no. 9644, 1:1,000 western blot), Phospho-4E-BP1 (Ser65) (catalogue no. 9456, 1:1,000 western blot), TFEB (catalogue no. 4240, 1:1,000 western blot) and Phospho-TFEB S211 (catalogue no. 37681, 1:1,000 western blot) were from Cell Signalling Technology; antibodies to GAPDH (6C5) (catalogue no. sc-32233, 1:15,000 western blot) and LAMP-1 (H4A3) (catalogue no. sc-20011, 1:500 immunofluorescence) were from Santa Cruz; antibody to HA.11 Epitope Tag (catalogue no. 901513) was from Biolegend and HRP-conjugated secondary

antibodies to mouse (catalogue no. 401215, 1:5,000 dilution) and rabbit (catalogue no. 401315, 1:5,000 dilution) IgGs were from Calbiochem.

Chemicals used were Torin1 (catalogue no. 4247) that came from Tocris, Protease Inhibitor Cocktail (catalogue no. P8340) and puromycin (catalogue no. P9620) were from Sigma-Aldrich and PhosSTOP phosphatase inhibitor cocktail tablets (catalogue no. 04906837001) were from Roche.

Plasmids used were the plasmid encoding full-length TFEB-GFP, previously described in ref. [9]. pRK5-HA-GST RagC wild-type (no. 19304) and pRK5-HA-GST-RagA wild-type (no. 19298) plasmids, which were a kind gift from D. Sabatini (Addgene plasmids). All the mutants used in these cellular assays were generated by using QuikChange II-E Site-Directed Mutagenesis Kit (no. 200555, Agilent Technologies).

### Cell culture

HeLa cells were cultured in MEM (catalogue no. ECB2071L, Euroclone) supplemented with 10% inactivated fetal bovine serum (FBS) (catalogue no. ECS0180L, Euroclone), 2 mM glutamine (catalogue no. ECB3000D, Euroclone), penicillin (100 IU ml$^{-1}$) and streptomycin (100 µg ml$^{-1}$) (catalogue no. ECB3001D, Euroclone) and maintained at 37 °C and 5% $CO_2$. RagC KO HeLa cells and RagA KO HeLa cells were previously generated and described in ref. [7]. Flp-In 293 T-REx cells (catalogue no. R78007 Thermo Fisher) were grown in DMEM (catalogue no. D6429 Sigma-Aldrich), supplemented with 10% (vol/vol) FBS (catalogue no. 10270 Thermo Fisher), 100 U ml$^{-1}$ penicillin and 100 µg ml$^{-1}$ streptomycin (catalogue no. P0781 Sigma-Aldrich), 100 µg ml$^{-1}$ Zeocin (catalogue no. ant-zn-5b InvivoGen, Toulouse, France) and 15 µg ml$^{-1}$ Blasticidin (ant-bl-5b InvivoGen). Cell lines were validated by morphological analysis and routinely tested for absence of mycoplasma.

### Cell treatment

For experiments involving amino acid starvation, cells were rinsed twice with PBS and incubated for 60 min (unless stated otherwise) in amino acid-free Roswell Park Memorial Institute medium (catalogue no. R9010-01, USBiological) supplemented with 10% dialysed FBS. Serum was dialysed against 1× PBS through 3,500 molecular weight cut-off dialysis tubing to ensure absence of contaminating amino acids. For amino acid refeeding, cells were restimulated for 30 min with 1× water-solubilized mix of essential (catalogue no. 11130036, Thermo Fisher Scientific) and non-essential (catalogue no. 11140035, Thermo Fisher Scientific) amino acids resuspended in amino acid-free Roswell Park Memorial Institute medium supplemented with 10% dialysed FBS, plus glutamine. Where reported, cells were incubated with 250 nM Torin1 during amino acid restimulation.

### Cell lysis and western blotting

Cells were rinsed once with PBS and lysed in ice-cold lysis buffer (250 mM NaCl, 1% Triton, 25mM HEPES pH 7.4) supplemented with protease and phosphatase inhibitors. Total lysates were passed ten times through a 25-gauge needle with syringe, kept at 4 °C for 10 min and then cleared by centrifugation in a microcentrifuge (14,000 rpm at 4 °C for 10 min). Protein concentration was measured by Bradford assay. Cell lysates were resolved by SDS–PAGE on 4–12% Bis-Tris gradient gels (catalogue no. NP0323PK2 NuPage, Thermo Fisher Scientific) and analysed by immunoblotting with the indicated primary antibodies.

### Confocal microscopy

Cells were grown on eight-well Lab-Tek II Chamber Slides, treated as indicated and fixed with 4% paraformaldehyde for 10 min at room temperature. Blocking was performed with 3% bovine serum albumin in PBS + 0.02% saponin for 1 h at room temperature. Immunostainings were performed on dilution of primary antibodies in blocking solution and overnight incubation at 4 °C, followed by three washes and secondary antibody incubation in blocking solution for 1 h at room temperature. After three more washes, coverslips were finally mounted

in VECTASHIELD mounting medium with 4,6-diamidino-2-phenylindole and analysed using LSM 800 or LSM 880+ Airyscan systems (Carl Zeiss), with a Plan-Apochromat ×63/1.4 NA M27 oil immersion objective using immersion oil (catalogue no. 518F, Carl Zeiss) at room temperature. The microscopes were operated on the ZEN 2013 software platform (Carl Zeiss). After calculation of processing for the airyscan, images were processed in the ImageJ v.1.47. Mander's colocalization coefficient was calculated using JACoP ImageJ Plugin[61].

## Immunoprecipitation

Flp-In 293 T-REx cells were transfected with wild-type or mutant TFEB-GFP. HeLa RagC KO cells or RagA KO cells grown on 10 cm culture dishes were transiently transfected with the different HA-GST-RagC mutants and wild-type Flag-RagA or HA-GST-RagA[H104D/D107R/E111R] and Flag-RagC, respectively, using Fugene HD (catalogue no. E2311, Promega). As a control, cell lines were transfected with STREP-HA-GFP (control beads lane on immunoprecipitation blots). The following day, cells were treated with 330 nM Torin1 (catalogue no. 4247, Tocris) for 1 h. Subsequently, cells were washed twice with ice-cold PBS and incubated with 1 mg ml$^{-1}$ dithiobis(succinimidyl propionate) crosslinker (catalogue no. 22586, Thermo Fisher Scientific) for 10 min at room temperature. The crosslinking reaction was quenched by adding Tris-HCl pH 8.5 to a final concentration of 100 mM, the cells rinsed again with ice-cold PBS and lysed in 25 mM HEPES pH 7.4, 250 mM NaCl, 1% Triton supplemented with protease and phosphatase inhibitors. For immunoprecipitations performed with TFEB as a bait, the lysis buffer also included 0.1% SDS and 2 mM EDTA. Lysates were passed five times through a 25-gauge needle with syringe and then cleared by centrifugation (14,000 rpm at 4 °C for 10 min). Lysates were then incubated with GFP trap magnetic agarose beads (no, gtma-20, ProteinTech Group, Inc) or with haemagglutinin beads (catalogue no. A2095, Sigma) at 4 °C for 2 h, washed with 40× the beads volume of lysis buffer, and eluted from the beads. Aliquots of the lysates and eluates were resolved by SDS–PAGE on 8, 10 or 15% SDS–PAGE gels and analysed by immunoblotting with the indicated primary antibodies. The emitter-coupled logic signal was detected and recorded with Fusion FX EDGE. Quantification of western blots was performed by calculating the intensity of the protein bands using the densitometry analysis function of ImageJ. Values were normalized to the respective control in each experiment.

## Reporting summary

Further information on research design is available in the Nature Portfolio Reporting Summary linked to this article.

## Data availability

Structural coordinates were deposited in the PDB with accession codes 7UX2 (Raptor–TFEB–Rag–Ragulator), 7UXC (asymmetric unit of mTORC1–TFEB–Rag–Ragulator) and 7UXH (mTORC1–TFEB–Rag–Ragulator). The cryo-EM density maps were deposited in the Electron Microscopy Data Bank with accession numbers EMD-26840 (Raptor–TFEB–Rag–Ragulator complex with Raptor mask), EMD-26842 (Raptor–TFEB–Rag–Ragulator complex with c-Rag–Ragulator mask), EMD-26843 (Raptor–TFEB–Rag–Ragulator complex with TFEB-nc-Rag–Ragulator mask), EMD-26844 (consensus refinement of the Raptor–TFEB–Rag–Ragulator complex), EMD-26846 (composite map of the Raptor–TFEB–Rag–Ragulator complex), EMD-26852 (consensus refinement of mTORC1–TFEB–Rag–Ragulator complex with C2 symmetry), EMD-26857 (symmetry expansion of the mTORC1–TFEB–Rag–Ragulator complex) and EMD-26861 (composite map of the mTORC1–TFEB–Rag–Ragulator complex).

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

**Acknowledgements** We thank S. Fromm and T. Stasyk for contributions to early stages of the project, J. Remis and D. Toso for cryo-EM facility support, A. Joiner for comments on the manuscript, and H. R. Shin, R. Zoncu and many members of the Zoncu and Hurley laboratories for discussions. This work was supported by Genentech as part of the Alliance for Therapies in Neuroscience and the National Institute of General Medical Sciences, National Institutes of Health, grant no. R01 GM111730 (J.H.H.), the Italian Telethon Foundation (to G.N. and A.B.), Associazione Italiana per la Ricerca sul Cancro A.I.R.C. (grant nos. MFAG-23538 to G.N. and IG-22103 and 5x1000-21051 to A.B.), MIUR (grant nos. PRIN 2017YF9FBS to G.N. and PRIN 2017E5L5P3 to A.B.) and the European Research Council H2020 AdG (grant no. LYSOSOMICS 694282 to A.B.).

**Author contributions** Z.C. expressed and purified proteins, reconstituted the protein complexes and carried out cryo-EM imaging and structural analysis. G.N., A.E. and J.M. carried out mutational analysis, cell culture and cell imaging and protein phosphorylation analysis. M.E.G.A. carried out immunoblotting. L.A.H., A.B. and J.H.H. supervised research. Z.C., G.N., A.B. and J.H.H. wrote the first draft of the manuscript. All authors contributed to editing and revising.

**Competing interests** J.H.H. is a cofounder and shareholder of Casma Therapeutics and receives research funding from Casma Therapeutics, Genentech and Hoffmann-La Roche. A.B. is a cofounder and shareholder of Casma Therapeutics and advisory board member of Next Generation Diagnostics, Avilar Therapeutics and Coave. The other authors declare no competing interests.

**Additional information**
**Correspondence and requests for materials** should be addressed to Andrea Ballabio or James H. Hurley.

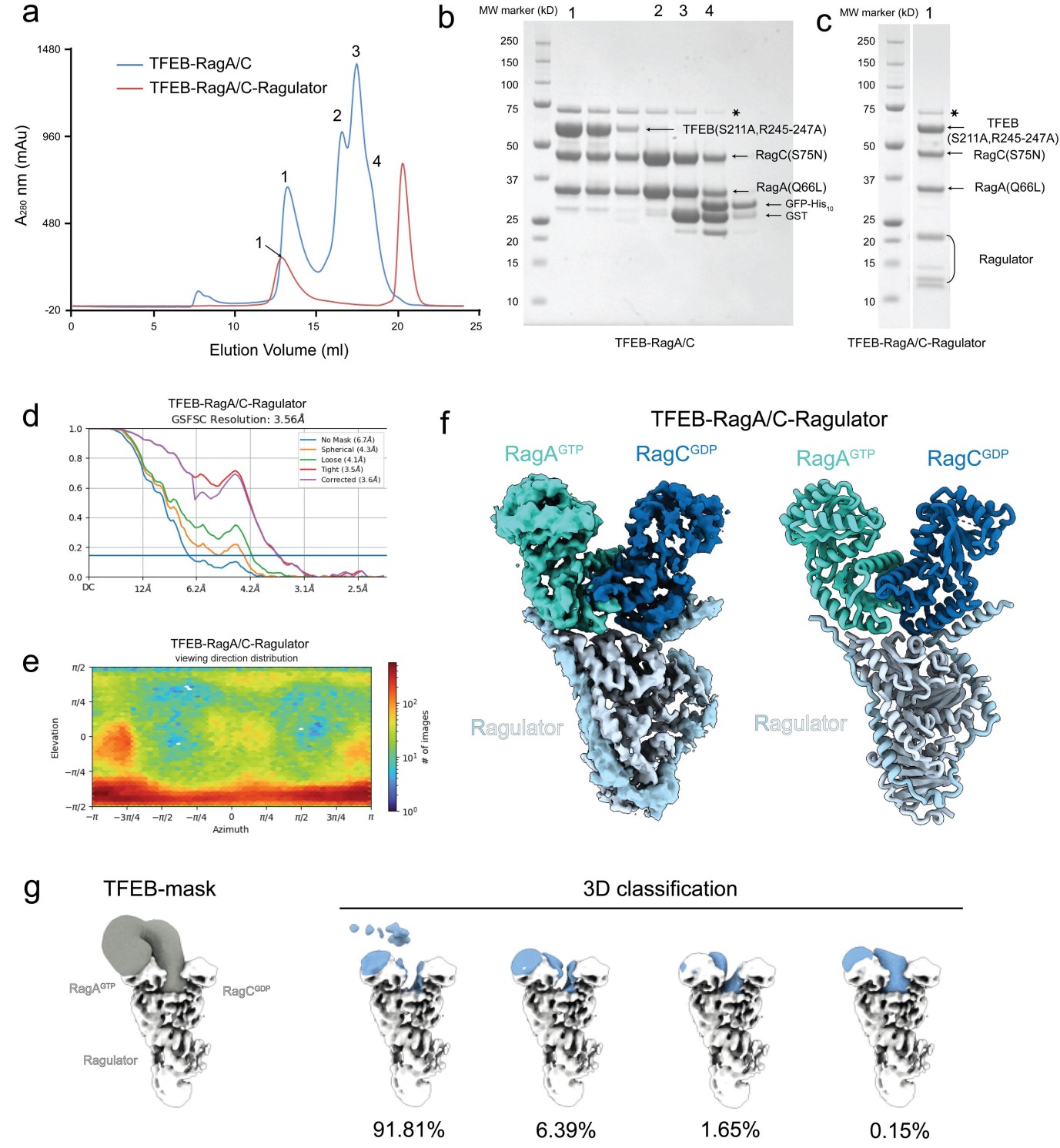

**Extended Data Fig. 1 | Purification, reconstitution, and cryo-EM structure determination of the TFEB-Rag-Ragulator complex. a**, Gel filtration chromatography of the TFEB-Rag and TFEB-Rag-Ragulator complexes, using a Superose 6 10/300 GL (GE Healthcare) column. Corresponding peaks are labelled and analyzed by Coomassie blue staining SDS-PAGE in **b** and **c** for TFEB-Rag and TFEB-Rag-Ragulator complexes, respectively. **d-f**, cryo-EM structure determination of TFEB-Rag-Ragulator complex. **d**, Resolution estimation based on the gold standard FSC. **e**, Orientation distribution of the reconstructed cryo-EM map. **f**, Side-by-side view of the cryo-EM density map and atomic model of the TFEB-Rag-Ragulator complex. Cryo-EM density for TFEB is not resolved. Asterisks in (**b**) and (**c**) indicate HSP70 contamination. **g**, Masked 3D classification of TFEB-Rag-Ragulator without alignment using the reconstruction in **f**. The TFEB mask is shown in grey density. The classification results are shown in blue. Rag-Ragulator is shown in light gray and shown as reference to identify TFEB. Only the least populated class (0.15%) shows density in the TFEB binding site.

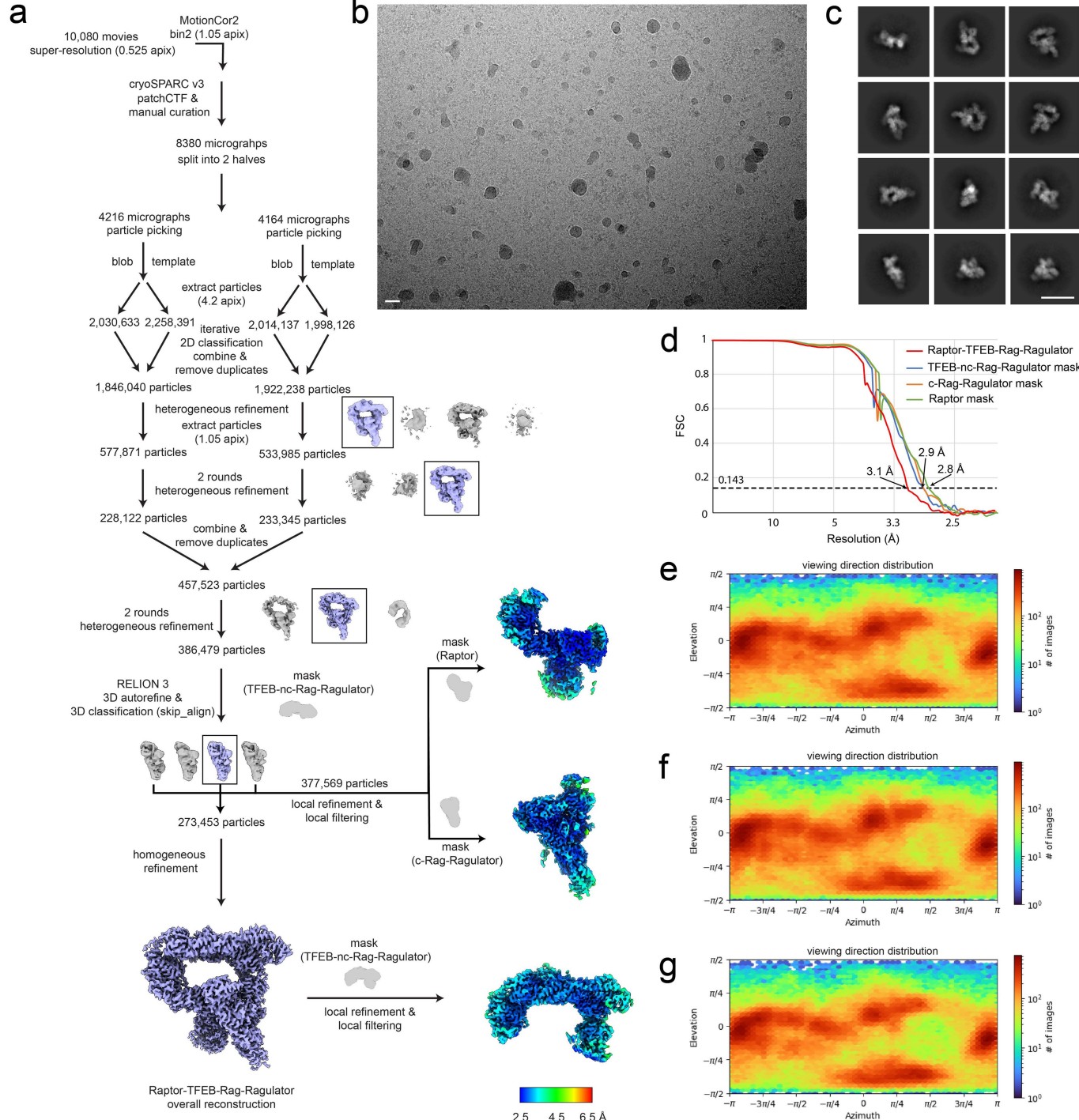

**Extended Data Fig. 2 | Cryo-EM workflow of the Raptor–TFEB-Rag-Ragulator complex. a**, Cryo-EM data processing diagram of the Raptor–TFEB-Rag-Ragulator complex. **b**, A representative micrograph of the dataset after motion correction. **c**, Selected 2D class average images showing different orientations of the complex. **d**, Resolution plots of the cryo-EM reconstructions with different masks. Orientation distribution of reconstructed cryo-EM maps with Raptor, c-Rag-Ragulator, and nc-Rag-Ragulator masks are shown in **e**, **f**, and **g**, respectively. Scale bars in **b** and **c** represent 20 nm.

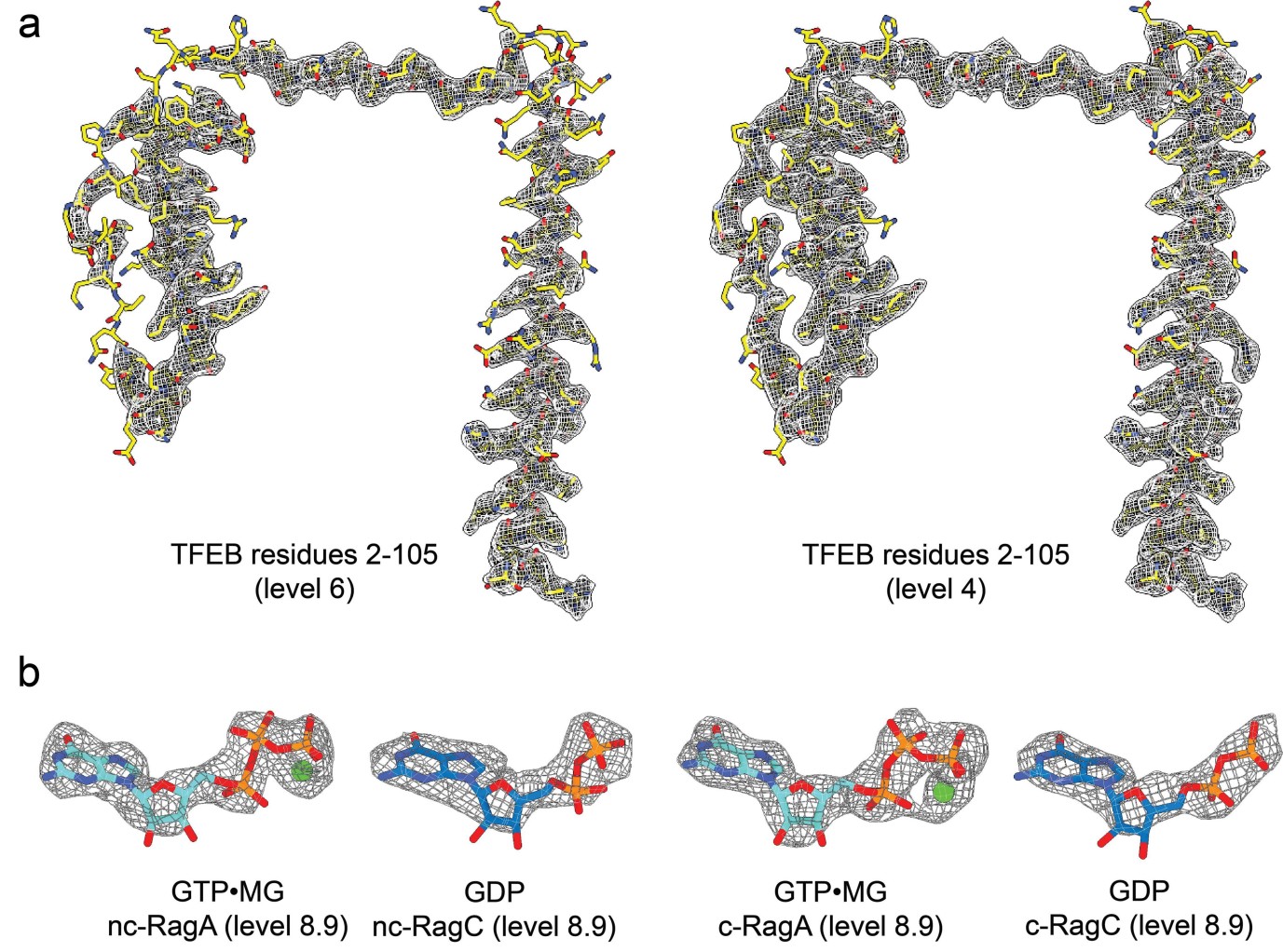

**a**

TFEB residues 2-105
(level 6)

TFEB residues 2-105
(level 4)

**b**

GTP•MG
nc-RagA (level 8.9)

GDP
nc-RagC (level 8.9)

GTP•MG
c-RagA (level 8.9)

GDP
c-RagC (level 8.9)

**Extended Data Fig. 3 | Representative cryo-EM density of the Raptor-TFEB-Rag-Ragulator complex. a**, Cryo-EM density of TFEB (2–105) at contour level 6 (left) and level 4 (right). **b**, Cryo-EM density of the nucleotides at contour level 8.9.

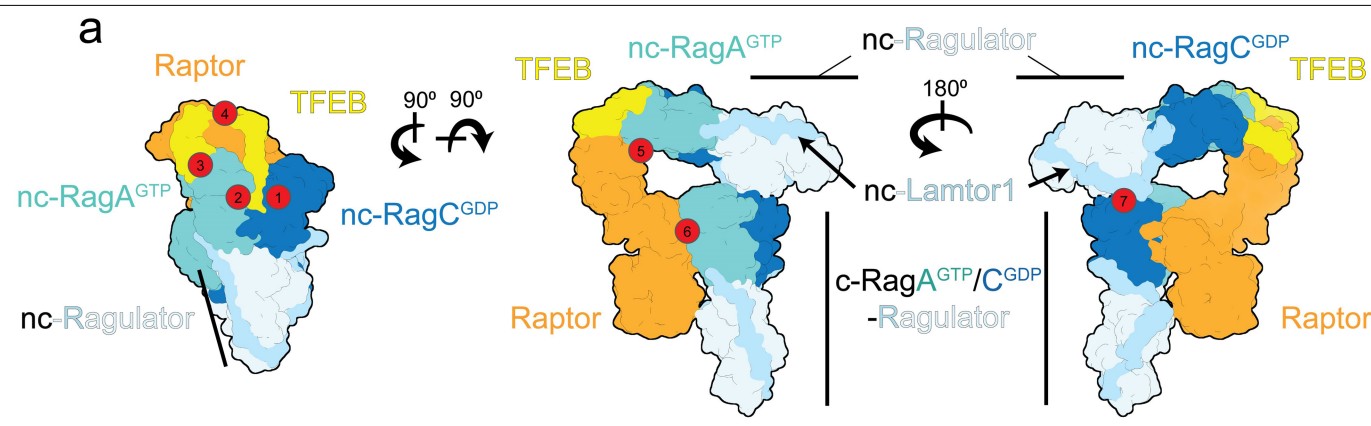

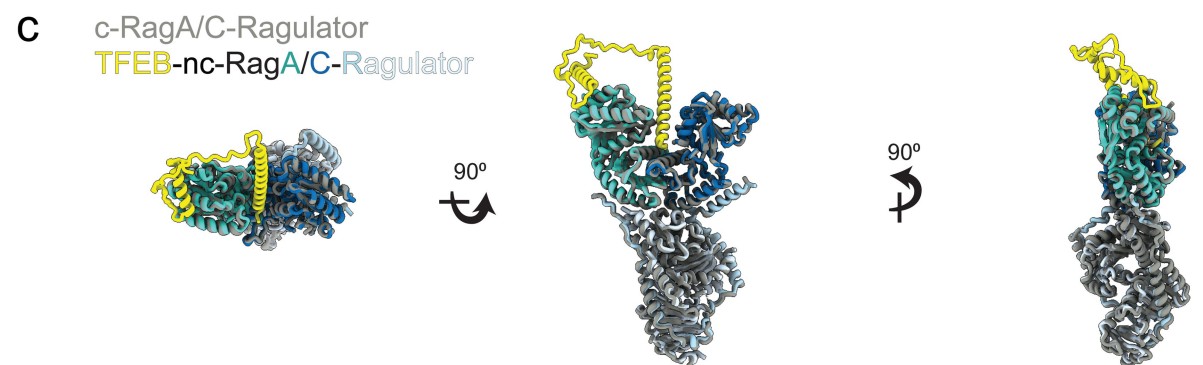

| Subunit 1 | Subunit 2 | Interaction interface Å² | |
|-----------|-----------|:---:|:---:|
| TFEB | nc-RagC<sup>GDP</sup> | ~570 | ① |
| | nc-RagA<sup>GTP</sup> (inter-Rag G domain cleft) | ~580 | ② |
| | nc-RagA<sup>GTP</sup> (outer-Rag G domain) | ~700 | ③ |
| | Raptor | ~1280 | ④ |
| Raptor | nc-RagA<sup>GTP</sup> | ~300 | ⑤ |
| | c-RagA<sup>GTP</sup> | ~1500 | ⑥ |
| nc-Lamtor1 | c-RagC<sup>GDP</sup> | ~380 | ⑦ |

**Extended Data Fig. 4 | Interaction surface aera estimation of the Raptor-TFEB-Rag-Ragulator complex and structural comparison between nc-Rag-Ragulator and c-Rag-Ragulator. a**, A cartoon representation of the complex, labelled with red circles indicating different interaction surface. **b**, A table showing the interaction surface aera labelled in **a**, estimated by PISA[62]. **c**, Structures are superimposed based on RagC. The TFEB-nc-Rag-Ragulator are colored as in Fig. 1. The c-Rag-Ragulator is colored in gray.

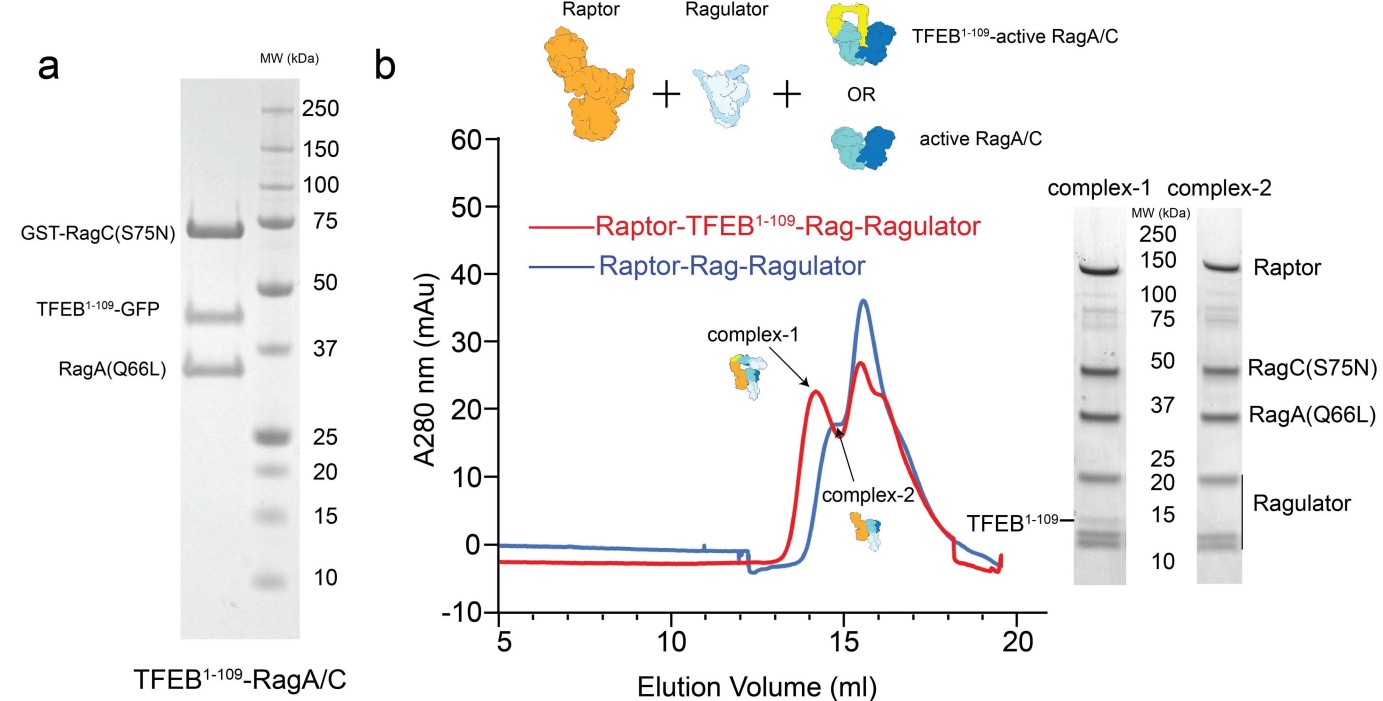

**Extended Data Fig. 5 | Assembly of Raptor-TFEB$^{1-109}$-Rag-Ragulator complex. a**, SDS-PAGE analysis of the purified TFEB$^{1-109}$-Rag complex before TEV cleavage and gel filtration. **b**, Gel filtration chromatography of the Raptor-Rag-Ragulator and Raptor-TFEB$^{1-109}$-Rag-Ragulator complexes, using a Superose 6 10/300 GL (GE Healthcare) column. The peaks corresponding to the largest complex are analyzed by SDS-PAGE and stained by Coomassie blue.

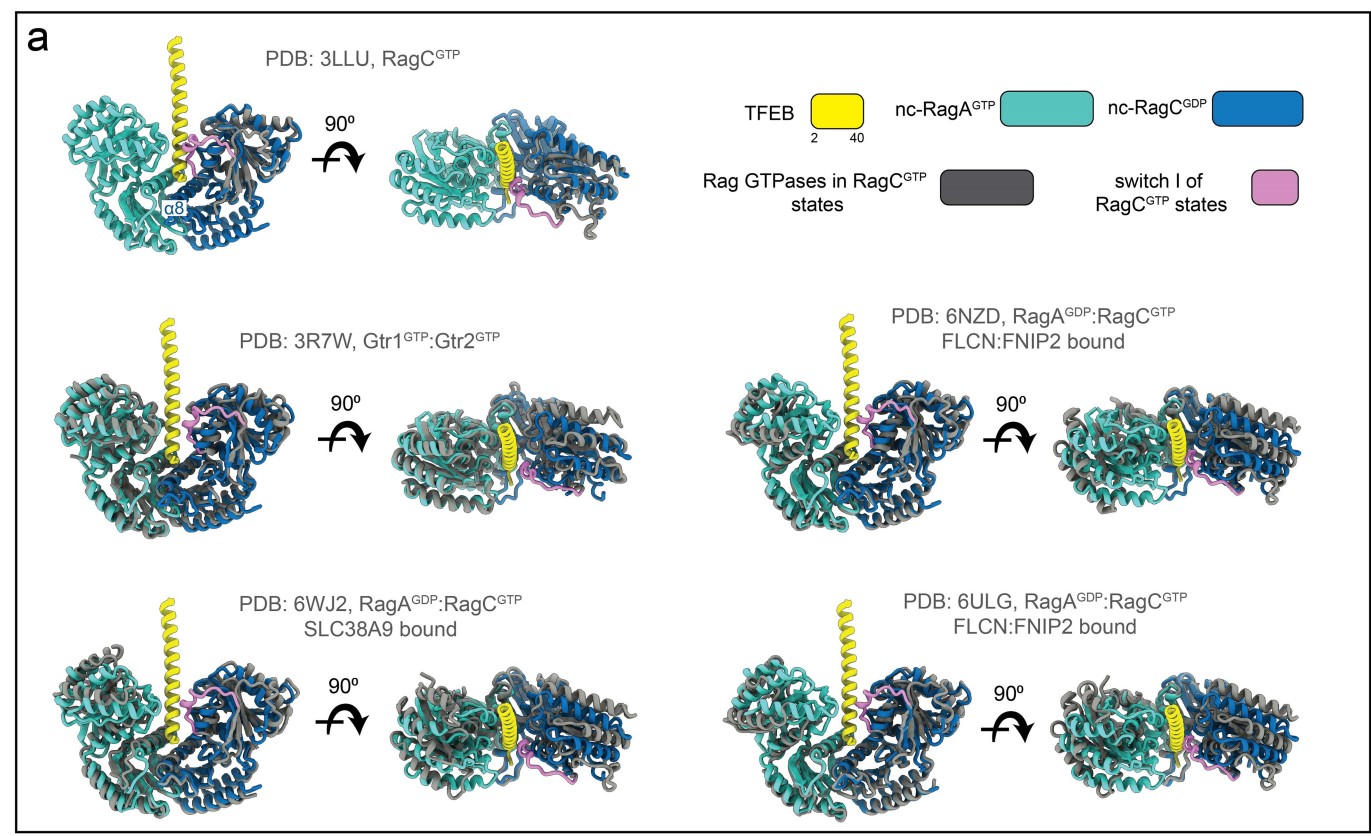

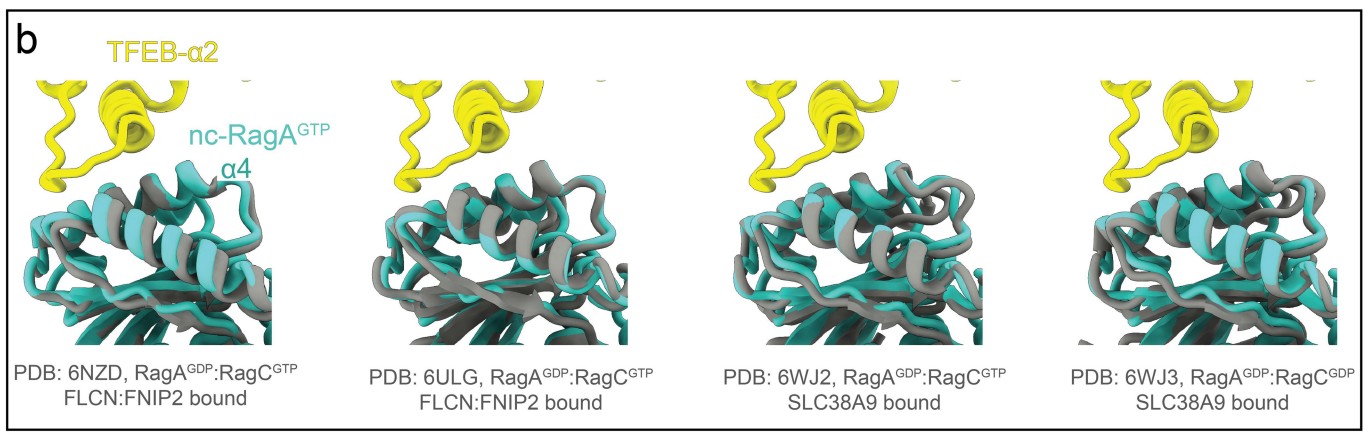

**Extended Data Fig. 6 | Structural comparison between the active nc-Rag GTPases (RagA$^{GTP}$-RagC$^{GDP}$) and RagC in GTP-bound states, and between the nc-RagA$^{GTP}$ and RagA in GDP-bound states at the unique TFEB contact site. a**, Structures are superimposed based on the α8 of RagC. Structures of RagC in GTP-bound states are colored as gray, while the switch I regions are colored as pink. Residues 41–105 of TFEB are omitted. **b**, Structures are superimposed based on the α4 of RagA. Structures of RagA at GDP-bound states are colored in gray. The TFEB and nc-Rag GTPases are colored as in Fig. 1.

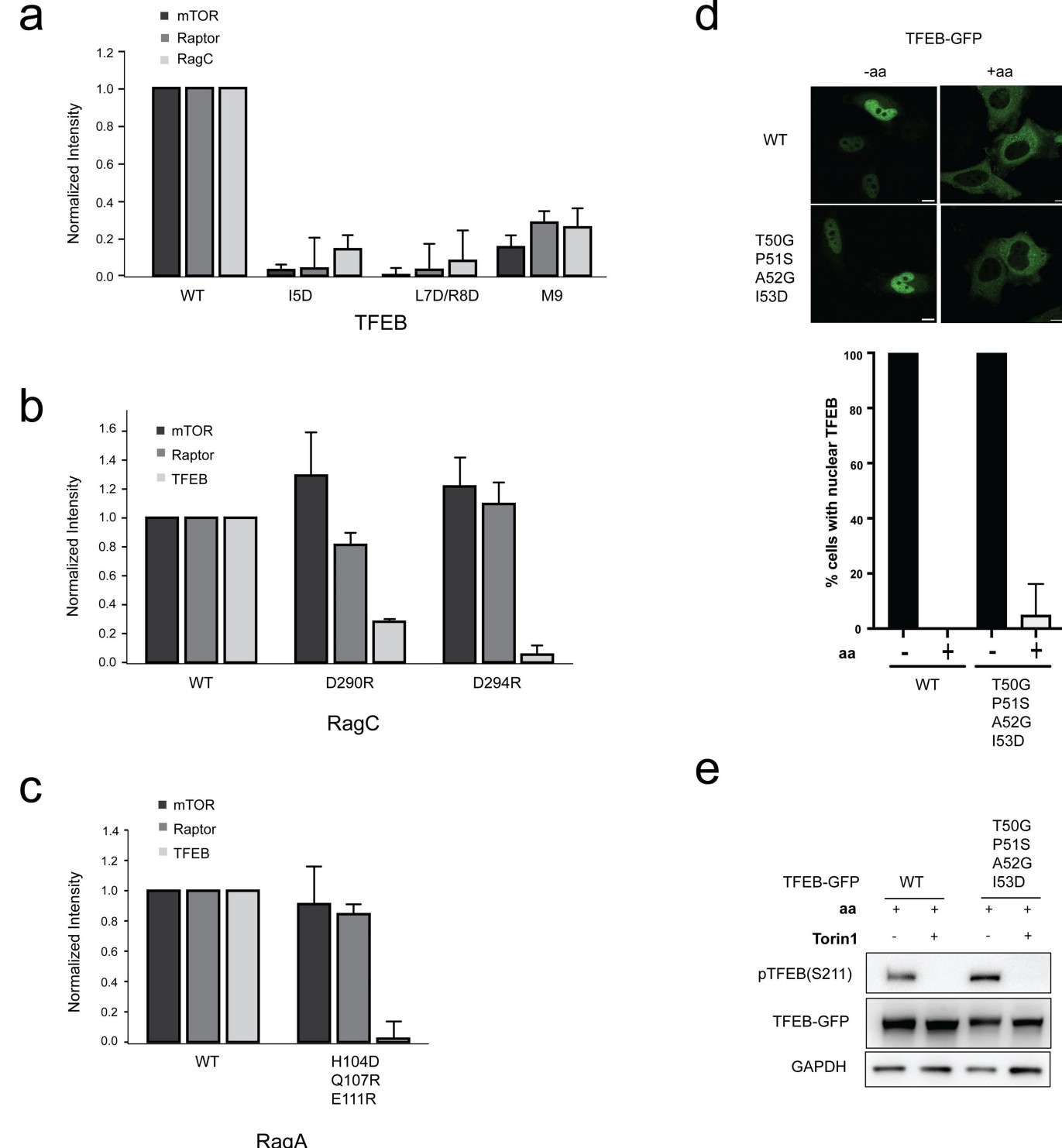

**Extended Data Fig. 7 | Quantification of the co-immunoprecipitation of mutants in the TFEB-nc-Rag GTPases interface and in cellulo assessment of the TFEB $^{50}$TPAI$^{53}$ mutation. a-c**, Quantifications of TFEB mutants in Fig. 3c, RagC mutants in Fig. 3e, and RagA mutants in Fig. 3g are calculated with mean ± s.e.m.; n = 3 experiments. **d**, Representative immunofluorescence analysis of cells expressing GFP tagged wild type or $^{50}$TPAI$^{53}$ mutant TFEB, in the presence and absence of amino acids. Quantification on the right shows the percentage of cells with TFEB nuclear localization. Results are mean ± s.e.; n = 5 independent fields per condition. Scale bar, 10 μm. **e**, TFEB phosphorylation was analyzed by immunoblotting for wild type and $^{50}$TPAI$^{53}$ mutant, in the presence and absence of Torin1.

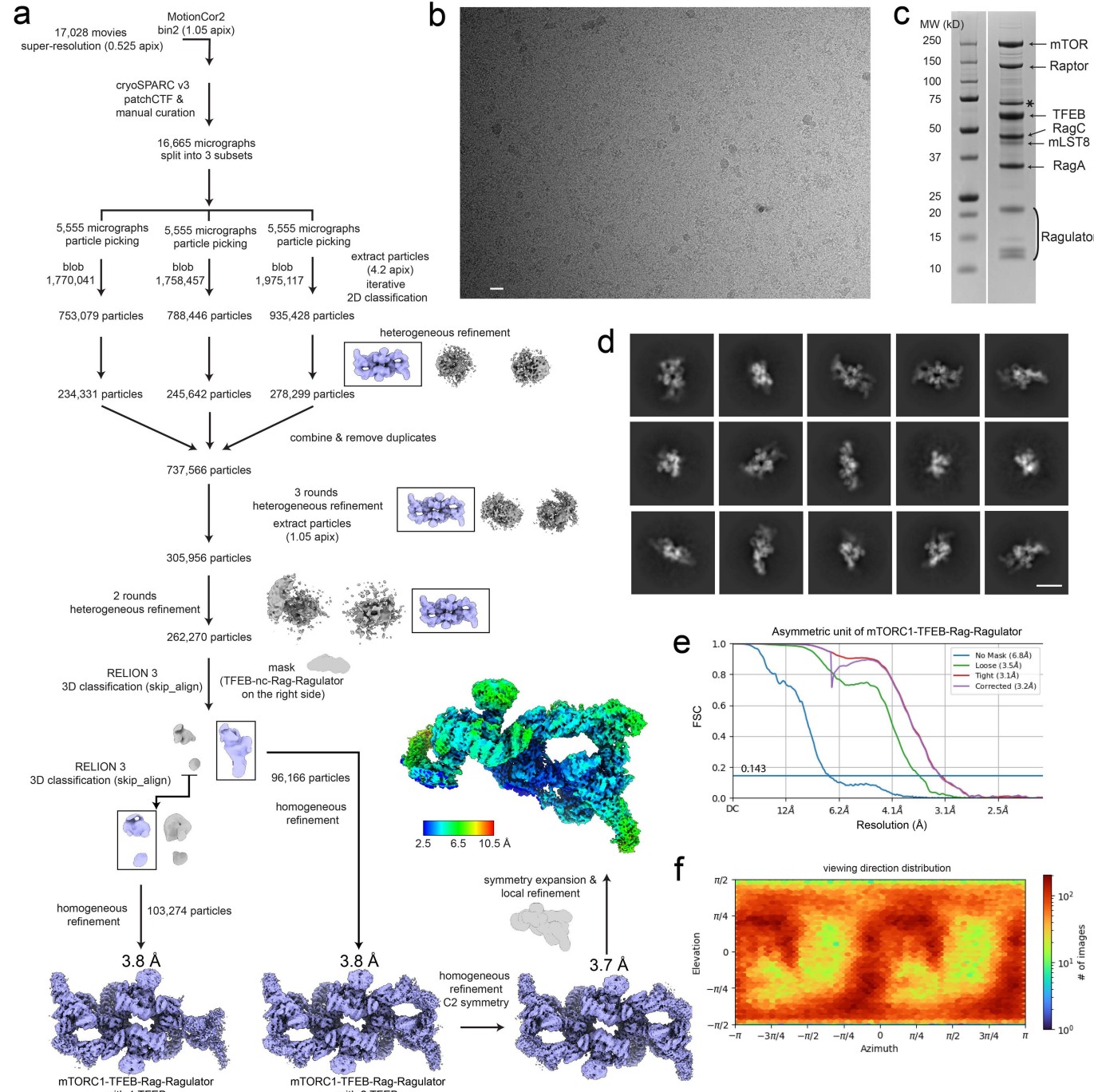

**Extended Data Fig. 8 | Cryo-EM workflow of the mTORC1-TFEB-Rag-Ragulator megacomplex. a**, Cryo-EM data processing diagram of the mTORC1-TFEB-Rag-Ragulator megacomplex. **b**, A representative micrograph of the dataset after motion correction. **c**, SDS-PAGE of the reconstituted megacomplex stained by Coomassie blue. Asterisk indicates HSP70

contamination. **d**, Selected 2D class average images showing different orientations of the complex. **e**, Resolution plots of the asymmetric unit of mTORC1-TFEB-Rag-Ragulator megacomplex after symmetry expansion. **f**, Orientation distribution of local refinement after symmetry expansion of the mTORC1-TFEB-Rag-Ragulator complex. Scale bars in **b** and **d** represent 20 nm.

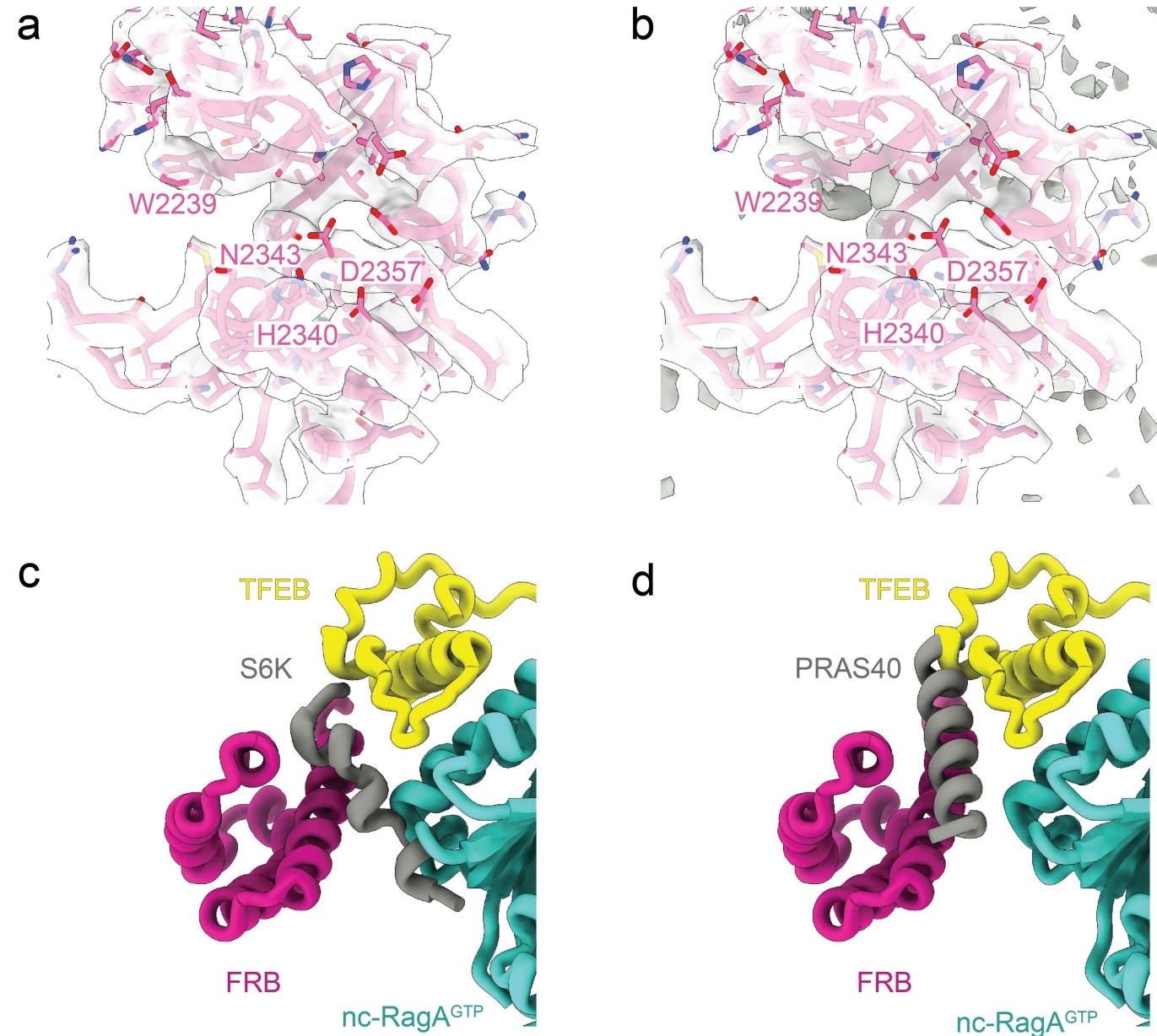

**Extended Data Fig. 9 | Cryo-EM map of the active site of mTOR and overlapping between TFEB, S6K and PRAS40. a**, Cryo-EM map in the active site of mTOR is shown as transparent surface. The density is zoned within 3Å of the model. **b**, The cryo-EM density outside the zone range is shown as solid surface colored in gray. **c-d**, Superimposed structure of S6K (PDB: 5WBH) and PRAS40 (PDB: 5WBU) with mTORC1-TFEB-Rag-Ragulator based on the FRB domain of mTOR.

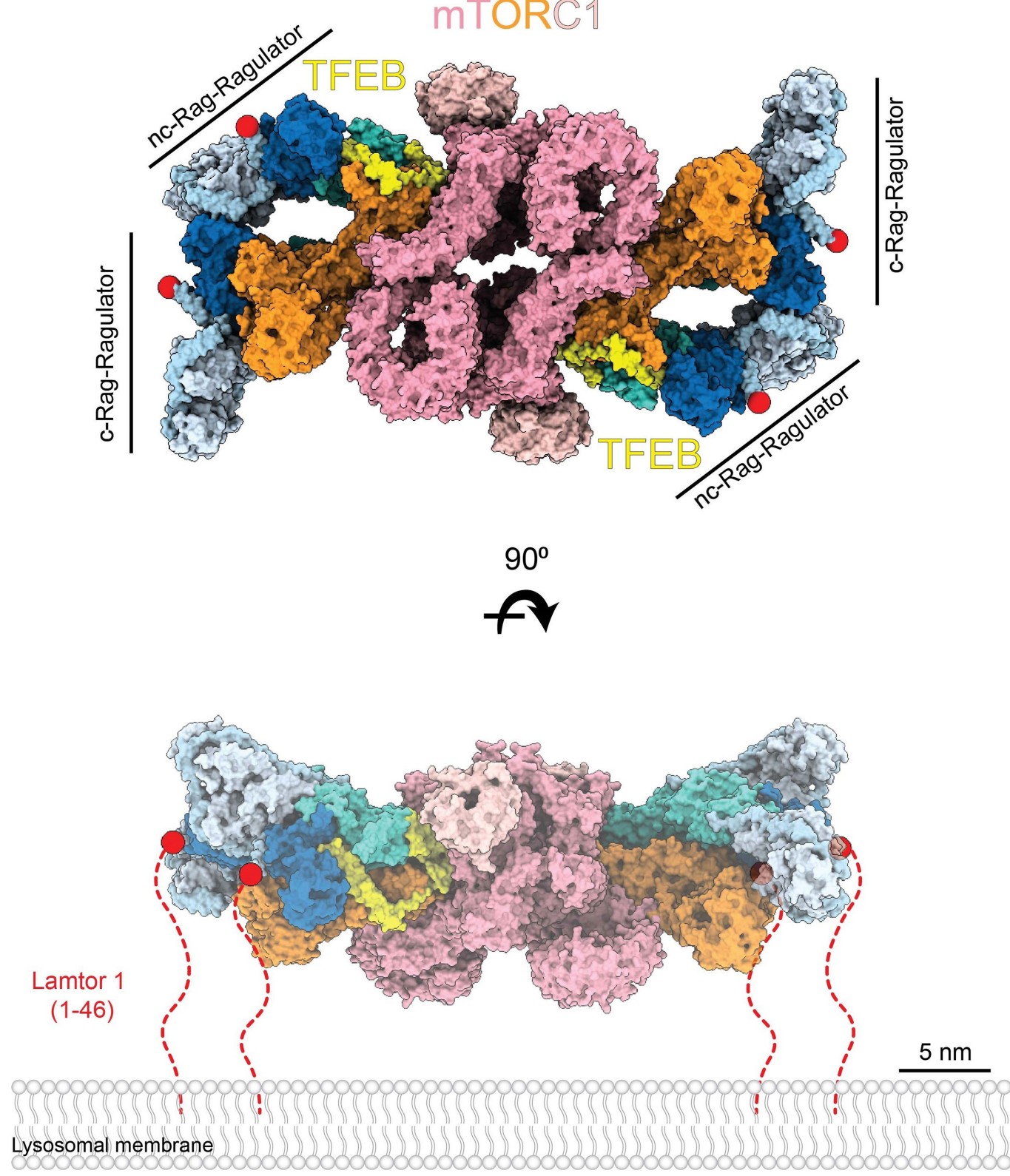

**Extended Data Fig. 10 | Proposed membrane tethering of the mTORC1-TFEB-Rag-Ragulator megacomplex.** The Red dots represent the position of residue 47 of Lamtor 1 subunit. The dashed curved lines indicate the disordered 46 residues of Lamtor1. The long linker of Lamtor 1 at N-terminus could also permit the complex facing toward the lysosomal membrane in an opposite orientation as shown above.

**Extended Data Table 1 | Cryo-EM data collection, refinement, and validation statistics**

|  | Raptor-TFEB-Rag-Ragulator (EMDB-26846) (PDB 7UX2) | mTORC1-TFEB-Rag-Ragulator (EMDB-26861) (PDB 7UXH) |
|---|---|---|
| **Data collection and processing** | | |
| Magnification | 81,000 | 81,000 |
| Voltage (kV) | 300 | 300 |
| Electron exposure (e–/Å$^2$) | ~50 | ~50 |
| Defocus range (μm) | -0.8 to -2.2 | -0.9 to -2.1 |
| Physical pixel size (Å) | 1.05 | 1.05 |
| Symmetry imposed | C1 | C2 |
| Initial particle images (no.) | 3,768,278 | 5,505,615 |
| Final particle images (no.) | 273,453 | 96,166 |
| Map resolution (Å) | 3.1 overall (2.8/2.9/2.9 local refine) | 3.2 (symmetry expansion) |
| FSC threshold | 0.143 | 0.143 |
| Map resolution range (Å) | ~2.6 to ~13.8 | ~2.6 to ~32 |
| | | |
| **Refinement** | | |
| Initial model used (PDB code) | 6U62 | 6BCX |
| Model resolution (Å) | 2.9 | 3.4 |
| FSC threshold | 0.5 | 0.5 |
| Map sharpening $B$ factor (Å$^2$) | -110.5 | -86.3 |
| Model composition | | |
| Non-hydrogen atoms | 27,264 | 95,194 |
| Protein residues | 3,421 | 11,918 |
| Ligands | 6 | 14 |
| $B$ factors (Å$^2$) | | |
| Protein | 34.80 | 101.71 |
| Ligand | 21.06 | 73.17 |
| R.m.s. deviations | | |
| Bond lengths (Å) | 0.003 | 0.005 |
| Bond angles (°) | 0.636 | 0.898 |
| Validation | | |
| MolProbity score | 1.57 | 1.28 |
| Clashscore | 4.37 | 1.81 |
| Poor rotamers (%) | 0.00 | 0.02 |
| Ramachandran plot | | |
| Favored (%) | 94.91 | 95.16 |
| Allowed (%) | 5.09 | 4.75 |
| Disallowed (%) | 0.00 | 0.09 |

Raptor-TFEB-Rag-Ragulator and mTORC1-TFEB-Rag-Ragulator models are refined using composite maps.

# Reporting Summary

## Statistics

For all statistical analyses, confirm that the following items are present in the figure legend, table legend, main text, or Methods section.

| n/a | Confirmed | |
|---|---|---|
| ☒ | ☐ | The exact sample size (*n*) for each experimental group/condition, given as a discrete number and unit of measurement |
| ☒ | ☐ | A statement on whether measurements were taken from distinct samples or whether the same sample was measured repeatedly |
| ☒ | ☐ | The statistical test(s) used AND whether they are one- or two-sided<br>*Only common tests should be described solely by name; describe more complex techniques in the Methods section.* |
| ☒ | ☐ | A description of all covariates tested |
| ☒ | ☐ | A description of any assumptions or corrections, such as tests of normality and adjustment for multiple comparisons |
| ☒ | ☐ | A full description of the statistical parameters including central tendency (e.g. means) or other basic estimates (e.g. regression coefficient) AND variation (e.g. standard deviation) or associated estimates of uncertainty (e.g. confidence intervals) |
| ☒ | ☐ | For null hypothesis testing, the test statistic (e.g. *F*, *t*, *r*) with confidence intervals, effect sizes, degrees of freedom and *P* value noted<br>*Give P values as exact values whenever suitable.* |
| ☒ | ☐ | For Bayesian analysis, information on the choice of priors and Markov chain Monte Carlo settings |
| ☒ | ☐ | For hierarchical and complex designs, identification of the appropriate level for tests and full reporting of outcomes |
| ☒ | ☐ | Estimates of effect sizes (e.g. Cohen's *d*, Pearson's *r*), indicating how they were calculated |

*Our web collection on statistics for biologists contains articles on many of the points above.*

## Software and code

Policy information about availability of computer code

| | |
|---|---|
| Data collection | SerialEM, MotionCor2, CryoSPARC v3, Relion3 |
| Data analysis | Relion 3, Chimera X, COOT, ISOLDE |

For manuscripts utilizing custom algorithms or software that are central to the research but not yet described in published literature, software must be made available to editors and reviewers. We strongly encourage code deposition in a community repository (e.g. GitHub). See the Nature Portfolio guidelines for submitting code & software for further information.

## Data

Policy information about availability of data

All manuscripts must include a data availability statement. This statement should provide the following information, where applicable:
- Accession codes, unique identifiers, or web links for publicly available datasets
- A description of any restrictions on data availability
- For clinical datasets or third party data, please ensure that the statement adheres to our policy

Structural coordinates have been deposited in the RCSB and EM density in EMDB and the accession codes have been provided. We have requested their release as soon as possible.

# Field-specific reporting

Please select the one below that is the best fit for your research. If you are not sure, read the appropriate sections before making your selection.

☒ Life sciences ☐ Behavioural & social sciences ☐ Ecological, evolutionary & environmental sciences

For a reference copy of the document with all sections, see nature.com/documents/nr-reporting-summary-flat.pdf

# Life sciences study design

All studies must disclose on these points even when the disclosure is negative.

| | |
|---|---|
| Sample size | N/A |
| Data exclusions | N/A |
| Replication | at least three replicates for cellular assays |
| Randomization | N/A |
| Blinding | N/A |

# Reporting for specific materials, systems and methods

We require information from authors about some types of materials, experimental systems and methods used in many studies. Here, indicate whether each material, system or method listed is relevant to your study. If you are not sure if a list item applies to your research, read the appropriate section before selecting a response.

## Materials & experimental systems

| n/a | Involved in the study |
|---|---|
| ☐ | ☒ Antibodies |
| ☐ | ☒ Eukaryotic cell lines |
| ☒ | ☐ Palaeontology and archaeology |
| ☒ | ☐ Animals and other organisms |
| ☒ | ☐ Human research participants |
| ☒ | ☐ Clinical data |
| ☒ | ☐ Dual use research of concern |

## Methods

| n/a | Involved in the study |
|---|---|
| ☒ | ☐ ChIP-seq |
| ☒ | ☐ Flow cytometry |
| ☒ | ☐ MRI-based neuroimaging |

## Antibodies

| | |
|---|---|
| Antibodies used | Antibodies to mTOR (Cat# 2983 - 1:100 IF) Phospho-p70 S6 Kinase (Thr389) (1A5) (Cat# 9206 - 1:1000 WB), p70 S6 Kinase (Cat# 9202 - 1:1000 WB), 4E-BP1 (Cat# 9644 - 1:1000 WB), Phospho-4E-BP1 (Ser65) (Cat# 9456 - 1:1000 WB), TFEB (Cat# 4240 - 1:1000 WB), Phospho-TFEB S211 (Cat# 37681 − 1:1000 WB) were from Cell Signaling Technology; antibodies to GAPDH (6C5) (Cat# sc-32233 - 1:15000 WB) and LAMP-1 (H4A3) (Cat# sc-20011 - 1:500 IF) were from Santa Cruz; antibody to HA.11 Epitope Tag (Cat# 901513) was from Biolegend; HRP-conjugated secondary antibodies to Mouse (Cat# 401215 - 1:5000 dilution) and Rabbit (Cat# 401315 - 1:5000 dilution) IgGs were form Calbiochem. |
| Validation | Manufacturer's websites, see above |

## Eukaryotic cell lines

Policy information about cell lines

| | |
|---|---|
| Cell line source(s) | HeLa |
| Authentication | Cell lines were validated by morphological analysis and routinely tested for absence of mycoplasma |
| Mycoplasma contamination | Cell lines were routinely tested for absence of mycoplasma. |
| Commonly misidentified lines (See ICLAC register) | N/A |

