## [Peer Review File · Nature]

Manuscript Title: Structure of the lysosomal mTORC1-TFEB-Rag-Ragulator megacomplex

Reviewer Comments & Author Rebuttals

Reviewer Reports on the Initial Version:

Referees' comments:

Referee #1 (Remarks to the Author):

In this manuscript, Cui et al. revealed structures of the mTORC1-Rags-Ragulator-TFEB complex and demonstrated how TFEB, an atypical substrate of mTORC1 (Rag-dependent but Rheb-independent), associates with the mTORC1-Rags-Ragulator complex. Interestingly, the cryo-EM structure revealed that TFEB directly interacts with GDP-RagC and Raptor, and the TFEB containing mTORC1-Rags-Ragulator subcomplex bears two Rags-Ragulator, in which two GDP-RagCs interact with Raptor through distinct interfaces; hence TFEB localizes at the interface of one of them. This mode of the Rags-Ragulator interaction with Raptor (mTORC1) has not been visualized previously, which is termed the "non-canonical Rags-Ragulator interaction" module in this paper. The studies identified many novel key residues within the interfaces of those components, including between non-canonical GDP-RagC and TFEB. In support of Rheb-independent mTORC1-induced TFEB phosphorylation, mutations of residues important for the interaction between TFEB and non-canonical GDP-RagC significantly reduced amino acid-induced TFEB phosphorylation while they neither affect lysosomal mTORC1 recruitment nor mTORC1-dependent other typical substrate phosphorylation, highlighting the importance of non-canonical conformation of the mTORC1-TFEB complex for specific phosphorylation of TFEB. Thus, this structural study explained why GDP-RagC is essential for mTORC1-dependent TFEB phosphorylation and why the TFEB phosphorylation by mTORC1 does not require Rheb.

The study's strengths represent identifying the unexpected novel mTORC1-TFEB-Rags-Ragulator complex with sophisticated structural data, contributing significantly to understanding the mechanistic basis of mTORC1-dependent TFEB regulation.

Weaknesses may include less physiological/biological breakthroughs in the mTOR or TFEB signaling, while the new structure explains mechanisms of the previously developed model for the atypical mode of mTORC1-dependent TFEB phosphorylation.

In addition, the study utilized ectopic overexpression of many mutant proteins, which might be necessary to stabilize/visualize the complex, but occasionally lead to a conclusion with undesirable artifacts or misleading.

From this reviewer's point of view, the authors may be able to add some biological/pathophysiological relevance to the newly identified non-canonical mTORC1-TFEB complex to further strengthen the paper.

1. In mammalian cells, are both endogenous canonical and non-canonical mTORC1 complexes coexist on the lysosome? If yes, what is the proportion of these mTORC1 complexes? This point can be studied utilizing any methods (e.g., lysosome purification, mass analyses, size exclusion chromatography using endogenously tagged TFEB or mTORC1 components).

2. Does any physiological or pathological conditions affect the formation of non-canonical mTORC1 complex bearing TFEB? Is there any transformation between canonical and non-canonical mTORC1 complex in response to nutrient availability or under disease conditions? For instance, does the constitutive active mTORC1 signaling (e.g., under loss of either TSC1 or TSC2) change the levels of TFEB containing non-canonical mTORC1?

3. Does the loss of TFEB expression affect the formation of the non-canonical mTORC1 complex?

4. Based on the authors' model, can RagD, a target of TFEB, also be in the non-canonical mTORC1 complex? please discuss this point.

Minor points

1. In Fig. 2i and 4c, it would be better to include pTFEB western blots. Levels of RagC expression are missing in Fig. 2i.

2. In Fig. 2j, k, and l, these observations should be verified through biochemical assays (e.g., co-immunoprecipitation)

3. In Fig. 4d, the two separately demonstrated WBs should be presented in a single gel as presented in Fig. 2h.

4. For Fig. 4g, biochemical assays (interaction between mTORC1 and RagC mutants) need to be included.

Referee #2 (Remarks to the Author):

Activation of the transcription factor EB (TFEB) upregulates autophagy and lysosomal degradation as key mechanisms for maintaining cellular homeostasis. Under growth-promoting nutrient conditions TFEB is phosphorylated by the master controller of growth, mTORC1, resulting in cytosolic retention and inactivation of TFEB. TFEB does not contain any of the well-characterized short linear motifs for mTORC1 substrate recruitment. Its phosphorylation by mTORC1 had been shown to be dependent on Rag GTPases, but mechanistic details remained elusive. Here, Cui and coworkers visualize complexes of TFEB, RAG GTPases, the lysosomal tethering complex Ragulator and mTOR and present a model for TFEB recruitment based on the simultaneous binding of two sets of activated RagA/RagC dimers in complex with Ragulator primarily to each Raptor component of dimeric mTORC1. Several structural observations are validated by in vitro biochemical analysis and fluorescent imaging and colocalization analysis in HEK293 cells.

The structural analysis matches community standards, is well documented and is presented at a detailed level. Some relevant aspects, in particular regarding the RagC interaction of TFEB, are convincingly validated while other aspects of the interactions observed or discussed here, e.g. the role of TFEB Raptor interaction or hypothesis on the active conformation of mTORC1 or the impact of membrane tethering on the complexes observed here, remain untested. Overall, the manuscript provides detailed and important novel insights into an mTORC1 recruitment mechanism unique to the substrate TFEB. It thus adds substantially to completing the picture of mTORC1 substrate interactions, with particular relevance to substrate-selective mTORC1 inhibition. For further comments, see below.

Comments:

L 46: "This structure presents the phosphorylatable Ser residues of TFEB to the mTORC1 active site in a suitable geometry for their phosphorylation." I suggest to remove this statement, as none of the serine residues is resolved, and "suitable geometry" would normally be understood as geometric match to a productive active site geometry, while here, at best, the extended linker may bridge the distance to the active site. The authors should discuss explicitly, whether any density, even at backbone level was observed in the active site, see comments below for further details.

L69: "Hierarchical phosphorylation of these sites": The authors mention hierarchical phosphorylation here, but don't further discuss how this might be achieved based on the binding mode observed in their work. Recent work has revealed contributions from recruitment mode and phosphorylation induced conformational changes for another substrate. The authors should discuss how the recruitment mode observed here may contribute to hierarchical phosphorylation, additional insights on how this is mediated at the substrate level would further strengthen the manuscript.

L97: The authors use a S211A mutant eliminating an mTORC1 phosphorylation site of TFEB for their structural work, because "it had been reported to stabilize TFEB association with Rags in cells". Mutants of S211A show constitutive nuclear localization in vivo, and interfere with 14-3-3 protein binding. While the latter aspect is irrelevant for in vitro studies, the authors should confirm that the S211A mutant of TFEB is still a substrate at all (at other sites) for mTORC1, and possibly that WT TFEB would bind in an identical manner. Apparently, the in vitro activity assays made for TFEB phosphorylation are targeting S211 phosphorylation. Any differences between WT and S211A may provide clues to hierarchical phosphorylation mechanisms and would impact conclusions on the active conformation of mTORC1 made later.

L102: It is surprising that no TFEB density at all was observed in the TFEB-Rag-Ragulator complex, despite it being stable in size exclusion chromatography. Did the authors confirm the absence even of spurious density after final analysis of all structures by classification? As this point is raised later again, validating the relevance of the Raptor-TFEB interface (absent here) would help to draw clearer conclusions.

L124: The pull-down for TFEB 1-109 might be substoichiometric. Could the authors confirm, e.g. in a competitive pulldown in the presence of wtTFEB that TFEB 1-109 or 2-105 has similar or identical

binding properties as wt TFEB, e.g. excluding additional binding sites when S211 is not mutated to A, e.g. to the kinase cleft?

L132: Salt bridges are not well visible, might be best to indicate them with dashed lines, e.g. for D290 and R4 or R8.

L140: Structurally, R4 appears to play a key role, but no results on mutants for this residue have been reported. If the authors have carried out mutational analysis for this residue, what are the results ?

L162: The authors reveal a previously unpredicted interaction of TFEB with RagA, which is not selective for the RagA GTP state, and the authors conclude that "RagA GTP is important for TFEB ... as for all known mTORC substrates": 1. Apparently TFEB makes unique interactions with RagA that other substrates don't make. Have the authors made any attempt to validate the relevance of this interaction for phosphorylation?

2. This conclusion would hint towards a relevant role of RagA for mTORC recruitment to the lysosomal membrane, which isn't discussed at all here; see also below.

L178: "Bridging two Rag-Ragulator complexes"; "bridge the Raptor N-terminal conserved (RNC) domain and the ordered switch I of nc-RagA": As the Raptor interactions (L178-L182) are not individually validated their role remains unclear. The validation of interactions mentioned in L186-L190 is not very clear and should be quantified (see comment on Fig. 4C below). The statement on the overall relevance of c-Rag to nc-Ragulator depends on this data.

Notably, the TFEB-Raptor interface is the largest individual protein-protein interface formed between two proteins subunits here and the interface area is in a range that may indicate a stable interaction. The authors should clarify whether TFEB already (stably) binds Raptor in the absence of RagA/Cs and Ragulator, and should discuss how other interactions of RagA/Cs with further characterized protein partners with overlapping binding sites might compete against the additional interactions observed here.

L188:" ..., Leu154, .." -> not indicated in Fig. 4b.

L200: " none of the RagC mutants was able to restore TFEB lysosomal localization" : Fig 4f shows TFEB-RagC colocalization, not TFEB lysosomal localization as in Fig. 2f.

L203: Blot in Fig. 4d should be quantified. It appears as Y150D/R198D binds considerably more strongly than Y150D ?

L216: C2 symmetry and symmetry expansion: Distribution of local resolution looks a little unusual with regions surrounding the mTOR kinase at local resolution around 6.5Å. C2 symmetry provides little improvement over C1 (0.1Å resolution). Occupancy of second TFEB-nc-RAG-Ragulator (with 2 TFEB) looks still weaker after classification without alignment. The authors might consider to analyze asymmetry and 3D conformational variability before applying C2 symmetry and to test different local refinement masks for optimizing local resolution. A possible benefit could be to obtain best quality visualization of the mTOR active site, to eventually confirm absence or presence of any

substrate density.

L230: Has the interaction been validated ? The indicated distances of 5-6Å would suggest the absence of direct interactions. Is this binding site of TFEB and FRB overlapping with other substrate-FRB binding sites ?

L234, 240 ff: Discussion of active site interaction/distance to active site: This point is relevant to any discussion of mTOR conformation and hierarchical phosphorylation. Did the authors observe any substrate or ligand density in the active site ? The active site density in any case should be represented in the paper. Could the use of nucleotides, or the Ser211A mutation have any impact on this ? Is the current positioning explaining any hierarchical mode of substrate phosphorylation ? (“conformational change in mTORC1 is unnecessary for TFEB phosphorylation”)

What is the indication that an active state turning over TFEB substrate sites was visualized ? Previous substrate complexes have been visualized with bound substrate, but empty kinase pocket, and likely didn't represent the enzymatically active state, which may have a low population in equilibrium. Making this statement would require resolving an active kinase trapped in action and possibly kinetic analysis of turnover rates under different conditions.

Discussion,

L269-274:

Relevance of raptor interface: Validating this would add to the overall relevance of this manuscript.

The discussion lacks any mentioning and reference to the membrane tethering of Ragulator via Lamtor1. Given the overall importance of membrane anchoring and lysosomal localization for mTORC1 activation, the authors should discuss the compatibility of their model with Lamtor1 membrane anchoring. Fig. 1 appears to indicate that Lamtor 1 is observed from residue 48 onwards with about 45 residues connecting to the membrane anchoring site. Which impact of membrane anchoring do the authors expect and could they visualize a plausible state of the full complex with all four Ragulator complexes anchored to the membrane ?

Methods

L320: Typo CHPAS

L341: Tagging of Raptor and mLST8: Could the authors more clearly specify which subunits were tagged at what ends ? Does tagging have any effect on the interactions discussed here as compared to pulled-down wild-type complexes ? Are any ordered tags or differences to other mTORC1 structural data observed ?

L347: Please provide buffer/gradient conditions for AEX.

Fig. 4c: There is a strong reduction in total TFEB for the double and triple mutants and changes in phosphorylation aren't easily apparent by eye. Please quantify blot with pTFEB relative to TFEB and 4EBP1 or similar standards, and for change in total TFEB.

Fig. 5: Consider to indicate membrane tethering of Ragulator for the overall megacomplex model (see also above).

Supplementary data

Table S1: More ligands than the bound GTP/GDP were observed, please mention in text and show density if relevant. B factors for protein and even more so for ligand are very low for the TFEB-Rag-Ragulator complex at around 3Å resolution, may well be fine, but please check.

Ext. Data Fig. 8: see comments above on symmetry application and local refinement.

Referee #3 (Remarks to the Author):

Summary:

mTOR is an evolutionarily conserved serine/threonine kinase that functions as the catalytic subunit of two multi-protein complexes: mTOR complex 1 (mTORC1) and mTOR complex 2 (mTORC2). mTORC1 plays a central role in regulating the production of proteins, lipids, and nucleotides and negatively suppressing catabolic processes including autophagy. mTORC1 consists of three core components: mTOR, Raptor, and mLST8. High-resolution cryo-EM have revealed the overall architecture and subunit organization of mTORC1 at high resolution as well as yielded insights into how regulators such as the Rag GTPases bind to this kinase complex. mTORC1 recognize several of its substrates through a 5-residue motif known as the TOS (TOR signaling) motif. In parallel, recent biochemical and NMR-based studies demonstrated how Raptor binds TOS motif to orient the canonical 4E-BP substrate towards the active site of mTOR in mTORC1. However, many mTORC1 substrates lack TOS motif and how these targets are recruited and presented to mTORC1 is not clear. Amongst these is TFEB, a transcription factor that regulates genes involved in autophagy, lysosome biogenesis, and lipid metabolism. Upon starvation, TFEB translocates to the nucleus to activate genes in the aforementioned processes. In nutrient rich conditions, TFEB is phosphorylated by mTORC1 at Ser122, Ser142, Ser211. More specifically, Ser211 phosphorylation mediates 14-3-3 binding and cytoplasmic sequestration. Recent studies showed that TFEB interacts with the Rab GTPases and that the GTPase activation protein (GAP) for RagC FLCN is required for TFEB phosphorylation. In this manuscript, Cui and colleagues used biochemical and cryo-EM based approaches to elucidate the mechanism of how TFEB is recruited and presented to mTORC1. They first showed that TFEB forms a stable complex with RagAGTP/RagCGDP and Ragulator as well as with the mTORC1 subunit Raptor by in vitro reconstitution. They used cryo-EM to determine the 3.1 Å structure of the TFEB-Raptor-Rag-Ragulator complex. This structure showed that there are two copies of Rag-Ragulator with respect to Raptor with one copy interacting with Raptor in a “canonical” fashion as previously observed in the cryo-EM structure of mTORC-Rag-Ragulator and the second copy of Rag-Ragulator engaging in interaction with Raptor via a “non-canonical” interface. TFEB (residues 2 to 105) was observed to interact with Raptor as well as the non-canonical Rag-Ragulator. In particular, the N-terminal region of TFEB is clamped between the G domains of RagAGTP and RagCGDP with specificity defined by the G domain of RagCGDP. Structure-guided mutagenesis in conjunction with TFEB localization and Western blotting analyses validated the importance of the RagCGDP-TFEB interface for TFEB phosphorylation and function. Furthermore, structural alignment showed the basis of why TFEB can only bind RagC in the GDP-bound but not

GTP-bound state. Interestingly, they also showed using a similar mutagenesis-based approach that the interaction between canonical Rag and non-canonical Ragulator to be important for TFEB phosphorylation. Lastly, they reconstituted a megacomplex composed of TFEB-Rag-Ragulator and mTORC1 and determined its cryo-EM structure at 3.2 Å resolution. This structure revealed that the stoichiometry and mode of interaction between TFEB to Raptor and Ragulator is preserved compared to the TFEB-Raptor-Rag-Ragulator complex. It also enabled the visualization of TFEB Ser 122 and Ser 142 (mTORC1 phosphorylation sites) which are located in close proximity to the active site of mTOR. Overall, this manuscript reports a key breakthrough in understanding how the master regulator of autophagy and lysosome biogenesis, TFEB is specifically recruited by RagCGDP to mTORC1 and reveals how Raptor assists in binding substrate without TOS motif. This work will have broad appeal, especially to investigators working in mTOR signaling, kinase regulation and substrate recognition, autophagy, and structural biology. That said, there are a number of open questions arising from the presented data that would require the authors to address before it is considered for publication. Some of these questions are related to potential limitations of the primarily cryo-EM-based structural studies.

Major comments:

1. The author's initial goal was to determine the structure of TFEB bound Rag-Ragulator complex. They succeeded in reconstituting a stoichiometric complex, suggesting that the interaction between TFEB and Rag-Ragulator is relatively stable. Yet, they could not resolve the density of TFEB in the resulting 3D EM density map. However, because very little information is provided in Extended data Fig 1 regarding the data processing scheme used, it is hard to determine if the inability to visualize TFEB is due solely to the dissociation of TFEB during vitrification or stringent criteria used in image processing. Furthermore, it is not clear if the authors have attempted to use approaches such as chemical crosslinking to stabilize this assembly for vitrification.
2. This research group has extensive experience in using HDX-MS to map interaction interfaces between subunits within protein complexes. Given that they were able to produce different complexes (eg. Rag-Ragulator) and individual subunits (eg. Raptor), HDX-MS could provide additional data to validate the interfaces observed from the cryo-EM structures.
3. The observation that two copies of Rag-Ragulator are present in the TFEB-Raptor-Rag-Ragulator is interesting and a major discovery of this work. It would be important to describe whether or not the canonical and non-canonical Rag-Ragulator in this assembly adopt similar structures.
4. Although the presence of Raptor enabled the authors to successfully visualize TFEB interaction with Rag-Ragulator by cryo-EM and the overall structure of this complex largely resembles the one observed in the context of the megacomplex with mTORC1, this finding leads to the open question regarding if the observed TFEB-Raptor-Rag-Ragulator is a physiologically relevant intermediate. Especially given the stability of the TFEB-Raptor-Rag-Ragulator assembly, one could envisage that TFEB loading would involve first formation of the TFEB-Raptor-Rag-Ragulator prior to generation of the megacomplex with mTOR and mLST8.
5. The authors assessed the effects of structure-guided mutants through examining the localization of TFEB as well as phosphorylation of S6K, 4EBP-1, and TFEB. It would also be interested to assess

the effects of these mutants using standard autophagy assays. Related to this, have the authors generated structure-guided mutants of Raptor and assess the effects of these mutations on TFEB phosphorylation and localization?

6. Rag-Ragulator plays a key role in localizing mTORC1 to the lysosomal membrane. However, there was no discussion on the effect of the second copy of Rag-Ragulator on mTORC1 membrane localization. Previous cryo-EM studies have generated a structural model regarding how mTORC1 is orientated when bound to Rag-Ragulator (Rogala et al. 2019; Anandapadamanaban et al. 2019). In this study, the observation that TFEB requires two copies of Rag-Ragulator to be recruited to mTORC1 creates a conundrum. More specifically, based on the orientation shown in Figure 5C, the canonical Rag-Ragulator would be positioned right on top of a membrane but the non-canonical Rag-Ragulator is projected in an opposite direction away from the membrane. Can the authors explain this confusing observation?

Author Rebuttals to Initial Comments:

Responses to referees' comments:

Referee #1 (Remarks to the Author):

We are glad that the reviewer appreciates the novelty and sophistication of the structural findings, and thank them for their time and insightful questions concerning physiology. We agree completely with the reviewer's perception that the strength of this structural biology study is that we have uncovered a novel and complex structural mechanism underpinning important and previously reported physiology.

1. In mammalian cells, are both endogenous canonical and non-canonical mTORC1 complexes coexist on the lysosome? If yes, what is the proportion of these mTORC1 complexes? This point can be studied utilizing any methods (e.g., lysosome purification, mass analyses, size exclusion chromatography using endogenously tagged TFEB or mTORC1 components).

As a broad comment in response to points 1-3, it is of the essence that the nc-Rag-Ragulator-TFEB and the megacomplex with mTORC1 are not stable entities under normal physiological conditions. This is clear from physiological evidence, in that mutation of the NLS and either mutation of S211 or torin treatment are needed to visualize TFEB bound to Rags and trapped on lysosomes (e.g. Martina *et al.*, 2013), and from the *in vitro* evidence that the nc and mega complexes only form in the presence of TFEB. Thus, the nc complex under the normal (as opposed to mutationally trapped) conditions of catalytic turnover by mTORC1 is a transient state that assembles only long enough for TFEB to be phosphorylated and then released. It is unlike stably assembled entities such as Ragulator or mTORC1 itself in this regard. We agree with reviewer 1 that in principle it would be valuable to measure the abundance of the nc complex under physiological (not trapped) conditions by biochemical means in the presence and absence of disease states, nutrients, and TFEB. However, given the transient nature of the complex, this type of experiment is inherently unfeasible.

We know that the complex can be stabilized experimentally by expression of the TFEB S211A/NLS mutant construct, however analysis of an artificially stabilized version of the complex would not address the reviewer's question about its proportionate abundance in physiology. The direct answer to the reviewer's question is that the amount of mTORC1 within an assembled nc-Rag-Ragulator-TFEB or megacomplex under normal physiological conditions is vanishingly low, because under normal conditions the complex forms transiently for each phosphorylation event.

2. Does any physiological or pathological conditions affect the formation of non-canonical mTORC1 complex bearing TFEB? Is there any transformation between canonical and non-canonical mTORC1 complex in response to nutrient availability or under disease conditions?

For instance, does the constitutive active mTORC1 signaling (e.g., under loss of either TSC1 or TSC2) change the levels of TFEB containing non-canonical mTORC1?

The megacomplex is profoundly relevant to both physiology and disease, as the entity responsible for MiT-TFE inactivation during amino acid abundance and whose formation is lost upon FLCN inactivation in Birt-Hogg-Dubé syndrome (Napolitano et al. *Nature*, 2020) and in Tuberous Sclerosis (Alesi et al. *Nature Communication*, 2021). In both conditions, TFEB is dephosphorylated and nuclear in spite of mTORC1 hyperactivity towards S6K and 4E-BP1. This is due to an impairment of RagC/D activity leading to disassembly and inactivation of the non-canonical mTORC1 complex. The visualization of the complex, whose existence we predicted on the basis of the unique TFEB phosphorylation phenotype of *FLCN* loss of function, is what makes this work so significant. As mentioned above, the megacomplex itself is transient, so the appropriate read-out for its functional relevance is the level of phosphorylated TFEB in cells, which we report on. Amino-acid starvation (as shown here) and *FLCN* GAP inactive mutations (see Jansen et al. *bioRxiv* 2022.06.28.498039), which result in dephosphorylated TFEB, control the formation of nc-mTORC1 complex by controlling levels of RagC^{GDP}. Presumably TFEB dephosphorylation under the loss of TSC1 or TSC2 must also be due to impaired nc-mTORC1 formation. The mechanism underlying TFEB dephosphorylation in TSC1/2 is still elusive and will be the focus of future studies that are beyond the scope of this manuscript.

3. Does the loss of TFEB expression affect the formation of the non-canonical mTORC1 complex?

The biochemical data, as shown in Fig. 1 and ED Fig. 6, show that TFEB is absolutely required for the formation of the nc-mTORC1 complex. This is consistent with previous structures of Raptor with only Rag-Ragulator (Rogala *et al.* 2019), which did not observe any trace of the non-canonical Rag-Ragulator in the absence of TFEB.

4. Based on the authors' model, can RagD, a target of TFEB, also be in the non-canonical mTORC1 complex? please discuss this point.

We thank the reviewer for raising this important point. As shown by Napolitano *et al.* (*Nature* 2020), RagD is able to rescue TFEB cytosolic relocation after refeeding in the RagC/D knockdown cells. All of the residues of RagC that contact TFEB, notably Asp290 and Asp294, are also present in RagD. Therefore, both functional and structural lines of evidence are consistent with the presence of RagD in the non-canonical mTORC1 complex. This point is now mentioned near the end of the first paragraph of the section on "Interactions between TFEB and active Rag GTPases".

Minor points

1. In Fig. 2i and 4c, it would be better to include pTFEB western blots. Levels of RagC expression are missing in Fig. 2i.

The p-TFEB antibody does not work well with endogenous TFEB. The phosphorylation of the endogenous TFEB is indicated as band shift when blotting for total endogenous TFEB, which is the standard method for analyzing phosphorylation of endogenous TFEB. We only include pTFEB western blots for TFEB-GFP exogenously expressed in cell lines.

Levels of RagC expression are now included in the Fig. 3f (previously Fig. 2i)

2. In Fig. 2j, k, and l, these observations should be verified through biochemical assays (e.g., co-immunoprecipitation)

We now have included the co-immunoprecipitation of RagC-TFEB interaction and RagC-mTORC1 interaction in Fig. 3e (previously Fig. 2h). We transiently transfected RagC mutants to the RagC KO cell line, instead of the TFEB-GFP stable cell line, to avoid the potential effects of the presence of endogenous wild type RagC and over-expressed TFEB-GFP.

3. In Fig. 4d, the two separately demonstrated WBs should be presented in a single gel as presented in Fig. 2h.

This has been done and the data are now shown in Fig. 4d.

4. For Fig. 4g, biochemical assays (interaction between mTORC1 and RagC mutants) need to be included.

This has also been done and the data are now shown in Fig. 4d. We used RagC KO cells as done also in response to point 2 above. The single and double mutants have no effect on mTOR binding, consistent with the imaging data in Fig. 4g. A roughly two-fold reduction in binding is seen in the triple mutant, which contrasts with the lack of effect on mTOR localization shown in Fig. 4d. We infer that subcellular localization of mTOR, which benefits from additional interactions with lysosomal membrane components, is more robust than interaction as assayed by co-IP. The reduction in TFEB binding is similar to that for mTOR, which contrasts with the almost complete loss of RagC co-localization shown in Fig. 4f. We interpret this as telling us that TFEB localization to lysosomes is completely dependent on the c-nc bridge, but formation of the nc complex alone is largely sufficient for co-IP, with only a modest loss in interaction seen for the most drastic (triple) mutation.

Referee #2 (Remarks to the Author):

We thank the reviewer for their time and appreciate their enthusiasm for the quality of the structural work and its validation.

Comments:

L 46: “This structure presents the phosphorylatable Ser residues of TFEB to the mTORC1 active site in a suitable geometry for their phosphorylation.” I suggest to remove this statement, as none of the serine residues is resolved, and “suitable geometry” would normally be understood as geometric match to a productive active site geometry, while here, at best, the extended linker may bridge the distance to the active site. The authors should discuss explicitly, whether any density, even at backbone level was observed in the active site, see comments below for further details.

We have deleted the sentence.

L69: “Hierarchical phosphorylation of these sites”: The authors mention hierarchical phosphorylation here, but don’t further discuss how this might be achieved based on the binding mode observed in their work. Recent work has revealed contributions from recruitment mode and phosphorylation induced conformational changes for another substrate. The authors should discuss how the recruitment mode observed here may contribute to hierarchical phosphorylation, additional insights on how this is mediated at the substrate level would further strengthen the manuscript.

Since none of the phosphorylation sites are visualized in this structure, it is difficult to draw any rigorous conclusions about hierarchical phosphorylation. We have deleted the word “hierarchical” from the manuscript (formerly it was only mentioned in the introduction.) However, we are enlightened by the recent paper by Böhm *et al.* on dynamic phosphorylation process of 4E-BP1 (“The dynamic mechanism of 4E-BP1 recognition and phosphorylation by mTORC1”) and added the further citation and discussion of the paper. See responses regarding L178.

L97: The authors use a S211A mutant eliminating an mTORC1 phosphorylation site of TFEB for their structural work, because “it had been reported to stabilize TFEB association with Rags in cells”. Mutants of S211A show constitutive nuclear localization in vivo, and interfere with 14-3-3 protein binding. While the latter aspect is irrelevant for in vitro studies, the authors should confirm that the S211A mutant of TFEB is still a substrate at all (at other sites) for mTORC1, and possibly that WT TFEB would bind in an identical manner. Apparently, the in vitro activity assays made for TFEB phosphorylation are targeting S211 phosphorylation. Any differences between WT and S211A may provide clues to hierarchical phosphorylation mechanisms and would impact conclusions on the active conformation of mTORC1 made later.

As shown by Napolitano *et al.* (Nature communications, 2018) (Fig 4b), S142 can still be phosphorylated in S211A background in vivo. We also performed in-vitro phosphorylation assay using TFEB 211A/NLS mutant. TFEB 211A/NLS can be phosphorylated by mTORC1 at S142 position as detected by western blot, using anti-phospho TFEB (Ser142) antibody (Sigma-Aldrich, ABE1971-I). The TFEB phosphorylation assay was done in 2 hours and 5 hours.

Fig R1. In-vitro phosphorylation assay of TFEB (S211A/NLS) by mTORC1. Different components in the reaction are indicated as above. Top two blots are with anti-phospho TFEB (Ser142) antibody after 2 hours or 5 hours reaction. The bottom blot is with wild-type TFEB antibody.

As for wild type TFEB, we couldn't co-express and purify it with active RagA/C in HEK293F cells, presumably due to its toxicity when highly over-expressed

Fig R2. Purification of TFEB (wild-type)-Rags
 SDS-PAGE analysis of the purification for TFEB (wild-type) with co-expressed active Rags is shown on the left. The right panel for purification of TFEB (S211A/NLS) with co-expressed active Rags is shown for comparison. The purification is done with GST tag before TEV cleavage.

L102: It is surprising that no TFEB density at all was observed in the TFEB-Rag-Ragulator complex, despite it being stable in size exclusion chromatography. Did the authors confirm the absence even of spurious density after final analysis of all structures by classification? As this point is raised later again, validating the relevance of the Raptor-TFEB interface (absent here) would help to draw clearer conclusions.

We continued the data processing of the TFEB-Rag-Ragulator, with further classification without alignment using TFEB mask. As shown in the figure below, the TFEB mask is colored in gray, and Rag-Ragulator density is shown as a reference for the position of TFEB. The densities after 3D classification are colored in light blue. A major class (class 1) with 91.81% particles was resolved with fragmented density. Although class 4 with only 0.15% particles showed some density in between RagA and RagC, we can't draw a definitive conclusion that it is TFEB due to

the limited number of particles in this class. Overall, it still seems that negligible density is present for TFEB in the absence of Raptor.

Fig R3. 3D classification of TFEB-Rags-Ragulator without alignment

The TFEB mask is shown in gray density. The classification results are shown in blue. The Rags-Ragulator are in light gray and shown as reference to identify TFEB.

L124: The pull-down for TFEB 1-109 might be substoichiometric. Could the authors confirm, e.g. in a competitive pulldown in the presence of wtTFEB that TFEB 1-109 or 2-105 has similar or identical binding properties as wt TFEB, e.g. excluding additional binding sites when S211 is not mutated to A, e.g. to the kinase cleft?

Extended Data Fig.6 (formerly Extended Data Fig. 5) showed the peak composition from the gel filtration, the complex peak has overlap with Rag-Ragulator peak, therefore the stoichiometry is biased. We have included the co-purification of TFEB1-109 with Rags in the Extended Data Fig.6a, which shows that they are stoichiometric.

L132: Salt bridges are not well visible, might be best to indicate them with dashed lines, e.g. for D290 and R4 or R8.

We have added salt bridges between D290 and R4 and R8 in Fig. 2c (previously Fig2.g).

L140: Structurally, R4 appears to play a key role, but no results on mutants for this residue have been reported. If the authors have carried out mutational analysis for this residue, what are the results ?

The TFEB R4 mutant was previously assessed by Martina *et al.* (JCB, 2013), which disrupted TFEB-Rags interaction and induced TFEB nuclear localization. The mutations we evaluated here were all novel sites.

L162: The authors reveal a previously unpredicted interaction of TFEB with RagA, which is not selective for the RagA GTP state, and the authors conclude that “RagA GTP is important for TFEB ... as for all known mTORC substrates”: 1. Apparently TFEB makes unique interactions with RagA that other substrates don't make. Have the authors made any attempt to validate the relevance of this interaction for phosphorylation?

2. This conclusion would hint towards a relevant role of RagA for mTORC recruitment to the lysosomal membrane, which isn't discussed at all here; see also below.

We thank the reviewer for this important suggestion. Indeed we used a RagA KO HeLa cell line in order to test RagA triple mutant (H104D/Q107R/E111R) at the TFEB interaction interface and found significant effects and therefore updated the entire Fig 2 and 3 and a paragraph to accommodate the new data.

“We then used a RagA KO cell line to validate whether the structural analysis faithfully reflects cell physiology. The interaction between TFEB and the transiently transfected Rags was significantly impaired in the RagA triple mutant (H104D/Q107R/E111R) in comparison to wild-type RagA (Fig. 3g). The expression of ^{RagA^{H104D/Q107R/E111R}} prevented TFEB phosphorylation in amino acid replete cells (Fig. 3h), but did not affect the phosphorylation of S6K and 4E-BP1. In addition, RagA^{H104D/Q107R/E111R} expression supported mTOR-RagA co-localization in amino acid replete conditions (Fig. 3n), but not cytosolic localization and RagA co-localization of TFEB in the absence and presence of Torin1, respectively (Fig. 3l, m). Therefore, the unique RagA-TFEB interaction is necessary for TFEB phosphorylation.”

With respect to lysosomal membrane recruitment, the major role of RagA-GTP is to bind Raptor and so recruit mTORC1 to the lysosome, in phosphorylation of both canonical and non-canonical substrates, as visualized in previous structures from the Williams and Sabatini labs (Rogala *et al.* 2019; Anandapadamanaban *et al.* 2019).

L178: “Bridging two Rag-Ragulator complexes”; “bridge the Raptor N-terminal conserved (RNC) domain and the ordered switch I of nc-RagA”: As the Raptor interactions (L178-L182) are not individually validated their role remains unclear. The validation of interactions mentioned in L186-L190 is not very clear and should be quantified (see comment on Fig. 4C below). The statement on the overall relevance of c-Rag to nc-Ragulator depends on this data.

Notably, the TFEB-Raptor interface is the largest individual protein-protein interface formed between two proteins subunits here and the interface area is in a range that may indicate a stable interaction. The authors should clarify whether TFEB already (stably) binds Raptor in the absence of RagA/Cs and Ragulator, and should discuss how other interactions of RagA/Cs with further characterized protein partners with overlapping binding sites might compete against the additional interactions observed here.

We carried out a detailed comparison to the findings of the recent paper by Böhm *et al.* (Mol Cell, 2021), and we noticed that the interaction between TFEB residues ⁵⁰TPAI⁵³ and Raptor (RNC domain) resembles the interaction between the RAIP motif of 4E-BP1 and Raptor. ⁵⁰TPAI⁵³

binding residues that also bind to 4E-BP1 RAIP are highlighted in boxes in Fig. 2h. Interestingly, Madhanagopal *et al.* (Science, 2019) showed in the Figure S4 that the RAIP-like motif of PRAS40 also interacts with Raptor in a similar manner, although this observation preceded the Böhm study and the point was not discussed in their manuscript. The high-resolution structure of TFEB-Raptor interaction interface presented here could serve as a template to shed light on the interaction between Raptor and the RAIP motif of 4E-BP1 or PRAS40. Böhm consider the RAIP motif secondary to the TOS motif in its interaction strength and importance in 4E-BP1 phosphorylation, and there are contradictory data on whether the RAIP motif is important for in vivo phosphorylation of 4E-BP1. In experiments carried out in response to reviewer comments, we find that mutation of the TFEB⁵⁰TPAI⁵³ sequence had no effect on TFEB phosphorylation and subcellular localization. Thus, the role of the TPAI-Raptor interaction seems secondary in importance to the direct interactions with the Rags. We added new sentences both in result and discussion to include this observation.

“The residues Thr⁵⁰, Pro⁵¹, Ala⁵², and Ile⁵³ of TFEB (TPAI) cover a patch on the RNC domain, formed by residues Pro⁷³, Pro¹⁵⁶, Glu¹⁶³, Trp¹⁶⁵, Tyr¹⁷⁴, Ile¹⁷⁵, and Pro¹⁷⁶ (Fig. 3e). The mTORC1 substrate 4E-BP1 contains an N-terminal RAIP sequence that binds to the same site on the RNC domain {Bohm, 2021 #121}. TFEB Ala⁵² and Ile⁵³ also interact with Raptor residues Tyr¹⁷⁴ and Ile¹⁷⁵ by b-sheet augmentation, as had been proposed for the 4E-BP1 RAIP:RNC interaction {Bohm, 2021 #121}. However, mutational disruption of the ⁵⁰TPAI⁵³ sequence had no effect on TFEB phosphorylation or subcellular localization (Extended Data Fig. 8).”

Extensive co-IP experiments have been conducted by many labs demonstrating that TFEB doesn't stably bind Raptor in the RagA/B deficient cells (Napolitano *et al.*, 2020, Fig. 2b) and in the presence of inactive Rags (Settembre *et al.*, 2012, Fig 4. G)(Vega-Rubin-de-Celis *et al.*, 2017 Fig 2.G). The fact that Δ30TFEB-GFP, which abolishes the interaction with Rags, does not co-IP with Raptor indicates that TFEB won't interact with raptor without Rags (Roczniak-Ferguson *et al.*, 2012, Fig 5E) (Napolitano *et al.*, 2020, Extended Data Fig. 5)(Hsu *et al.*, 2018, Fig. 4).

We have added this text to the discussion:

“The presence of direct TFEB-Raptor interactions was completely unexpected. An Ile- and Pro-containing TPAI sequence of TFEB binds directly to a surface patch on the RNC domain of Raptor that was previously shown to bind the RAIP sequence of 4E-BP1 {Bohm, 2021 #121}. The TOS-containing mTORC1 inhibitor PRAS40 also contains an RAIP-like motif that interacts with the same patch on the RNC domain {Anandapadamanaban, 2019 #40}, although its functional role has not been assessed. We found that a quadruple TPAI mutant of TFEB has no effect on TFEB phosphorylation by mTORC1 and subcellular localization. This is consistent with the finding that TFEB is not stably bound to Raptor in RagA/B deficient cells {Napolitano, 2020 #12}, in the presence of inactive Rags {Settembre, 2012 #5}{Vega-Rubin-de-Celis, 2017 #95} or when the Rag-binding N-terminal residues of TFEB are absent {Roczniak-Ferguson, 2012 #82}{Napolitano, 2020 #12}{Hsu, 2018 #122}. The 4E-BP1 RAIP motif is separated from the TOS motif by 100 residues, and the RAIP-Raptor interaction makes a secondary contribution to

mTORC1 binding relative to the TOS motif. Our observations are consistent with at most a secondary role of the TPAI motif of TFEB in driving megacomplex assembly.”

For example, further characterized protein partners that interact with RagA/C include FLCN-FNIP2 and GATOR1, which promote active or in active RagA/C respectively. Although FLCN-FNIP2 shares overlapping binding sites with c-RagC and nc-Ragulator interface and GATOR1 shares overlapping binding sites with TFEB-Raptor-RagA interface, they are in the upstream of TFEB phosphorylation, which is under tight regulation and unlikely to compete with TFEB in the physiological condition.

We also added a paragraph in the discussion “The inter-cleft binding site for TFEB in the nc-Rag dimer overlaps with the site occupied by FLCN-FNIP2 in the LFC {Lawrence, 2019 #66} and with SLC38A9 {Fromm, 2020 #91}, highlighting the complex time-sharing of different regulatory factors in the cleft.”

L188:” ..., Leu154, ..” -> not indicated in Fig. 4b.

We have added Leu154 to Fig 4b.

L200: “ none of the RagC mutants was able to restore TFEB lysosomal localization” : Fig 4f shows TFEB-RagC colocalization, not TFEB lysosomal localization as in Fig. 2f.

We have changed the sentence:

“...none of the RagC mutants were able to restore TFEB-RagC co-localization”

L203: Blot in Fig. 4d should be quantified. It appears as Y150D/R198D binds considerably more strongly than Y150D ?

The quantification has been added to Fig. 4d. Neither the single nor double mutants perturb binding to any great degree.

L216: C2 symmetry and symmetry expansion: Distribution of local resolution looks a little unusual with regions surrounding the mTOR kinase at local resolution around 6.5Å. C2 symmetry provides little improvement over C1 (0.1Å resolution). Occupancy of second TFEB-nc-RAG-Ragulator (with 2 TFEB) looks still weaker after classification without alignment. The authors might consider to analyze asymmetry and 3D conformational variability before applying C2 symmetry and to test different local refinement masks for optimizing local resolution. A possible benefit could be to obtain best quality visualization of the mTOR active site, to eventually confirm absence or presence of any substrate density.

We performed 3D classification before applying C2 symmetry expansion and found a small degree of 3D variability, shown in panel a of the figure below. We therefore did focused

refinement with mTOR-Kinase domain mask, TFEB-Rag-Ragulator mask and Raptor mask indicated in the panel b. The results are shown in the figure below. 1, the active site from the focused refinement using mTOR-Kinase domain mask (light blue) is still empty. 2, side chain resolvability of TFEB density with TFEB-Rag-Ragulator mask is less clear than the density with symmetry expansion (purple density). 3. The density for Raptor using Raptor mask (yellow) is similar quality to the map from symmetry expansion reconstruction (purple). Nevertheless, reconstructions of mTORC1-TFEB-Rag-Ragulator complex with different mask without using C2 symmetry expansion did not yield better local resolution. In addition, the active site of mTOR still remains empty.

Fig R4. 3D classification of Mtorc1-TFEB-Rag-Ragulator megastructure before symmetry expansion

a, overlay of densities after 3D classification without applying symmetry. b, Different masks are indicated for focused refinement. The light blue mask covers the kinase domain. The TFEB-Rag-Ragulator and Raptor masks are shown in gray and yellow, respectively. Densities of focused refinement results are as labeled: 1. Active site of mTOR; 2. TFEB-Rag-Ragulator (gray) 3. Raptor. The purple density is the after applying symmetry expansion and shown for comparison.

L230: Has the interaction been validated ? The indicated distances of 5-6Å would suggest the absence of direct interactions. Is this binding site of TFEB and FRB overlapping with other substrate-FRB binding sites ?

We have changed sentence to:

“In addition, Tyr95 and Thr99 of TFEB are within 5 Å of, but do not directly contact, a hinge loop (residues 2115-2118) at the end of mTOR FRB domain (Fig. 5d)”.

We added a new Extended Data Fig. 10, showing the relative positions of S6K, PRAS40 and TFEB, and a sentence in the manuscript:

“... the kinase domain (KD) N lobe and C lobe (Fig. 5c). TFEB shows limited overlap with the PRAS40 {Yang, 2017 #89}, while no overlap with the S6K, at the FRB binding site (Extended Data Fig. 10c, d).”

L234, 240 ff: Discussion of active site interaction/distance to active site: This point is relevant to any discussion of mTOR conformation and hierarchical phosphorylation. Did the authors observe any substrate or ligand density in the active site? The active site density in any case should be represented in the paper. Could the use of nucleotides, or the Ser211A mutation have any impact on this? Is the current positioning explaining any hierarchical mode of substrate phosphorylation?

(“conformational change in mTORC1 is unnecessary for TFEB phosphorylation”)

What is the indication that an active state turning over TFEB substrate sites was visualized? Previous substrate complexes have been visualized with bound substrate, but empty kinase pocket, and likely didn't represent the enzymatically active state, which may have a low population in equilibrium. Making this statement would require resolving an active kinase trapped in action and possibly kinetic analysis of turnover rates under different conditions.

We agree that it is common to have bound substrate but with empty kinase pocket, which may be due to the low population in equilibrium. We have added the cryo-EM density of the active site in Extended Data Fig. 10a, b, and edited the manuscript:

“Residues 2-108 of TFEB were resolved in the cryo-EM structure, essentially as before, but with the rest of TFEB not visualized and an empty active site despite the presence of a non-hydrolyzable ATP analogue and the presence of sequences containing phosphorylation sites (Extended Data Fig. 10a, b).”

We have deleted the word “hierarchical” from the manuscript (formerly it was only mentioned in the introduction) due to phosphorylation sites not being visualized in this structure. And we also modified the sentence (“conformational change in mTORC1 is unnecessary for TFEB phosphorylation”) to the following:

“Our results shed light on the alternative structural mechanism of Rheb-independent TFEB phosphorylation by mTORC1, with.....”

The experiments suggested by the reviewer will be interesting to pursue in the future as a way to trap complexes with ordered phosphorylatable or substrate-mimetic binding modes.

Discussion,

L269-274:

Relevance of raptor interface: Validating this would add to the overall relevance of this manuscript.

As shown the new Extended Data Fig. 8, mutation of the four residues in the TFEB TPAI sequence most centrally involved in this interface has no effect on function. More drastic mutants led to a loss of TFEB expression. Raptor cannot be completely knocked down because of its essential role in cell proliferation, therefore, mutations in the TFEB binding site of Raptor cannot be studied by knockdown/rescue experiments. On the basis of the lack of phenotype upon mutation of the TPAI motif, and as discussed above in reference to L178, it appears this interaction is secondary to the contacts with the Rags in its contribution to function.

We have added these sentences in the discussion:

“The presence of direct TFEB-Raptor interactions was completely unexpected. An Ile- and Pro-containing TPAI sequence of TFEB binds directly to a surface patch on the RNC domain of Raptor that was previously shown to bind the RAIP sequence of 4E-BP1 {Bohm, 2021 #121}. The TOS-containing mTORC1 inhibitor PRAS40 also contains an RAIP-like motif that interacts with the same patch on the RNC domain {Anandapadamanaban, 2019 #40}, although its functional role has not been assessed. We found that a quadruple TPAI mutant of TFEB has no effect on TFEB phosphorylation by mTORC1 and subcellular localization. This is consistent with the finding that TFEB is not stably bound to Raptor in RagA/B deficient cells {Napolitano, 2020 #12}, in the presence of inactive Rags {Settembre, 2012 #5}{Vega-Rubin-de-Celis, 2017 #95}, or when the Rag-binding N-terminal residues of TFEB are absent {Roczniak-Ferguson, 2012 #82}{Napolitano, 2020 #12}{Hsu, 2018 #122}. The 4E-BP1 RAIP motif is separated from the TOS motif by 100 residues, and the RAIP-Raptor interaction makes a secondary contribution to mTORC1 binding relative to the TOS motif. Our observations are consistent with at most a secondary role of the TPAI motif of TFEB in driving the megacomplex assembly.”

The discussion lacks any mentioning and reference to the membrane tethering of Ragulator via Lamtor1. Given the overall importance of membrane anchoring and lysosomal localization for mTORC1 activation, the authors should discuss the compatibility of their model with Lamtor1 membrane anchoring. Fig. 1 appears to indicate that Lamtor 1 is observed from residue 48 onwards with about 45 residues connecting to the membrane anchoring site. Which impact of membrane anchoring do the authors expect and could they visualize a plausible state of the full complex with all four Ragulator complexes anchored to the membrane ?

We have included an Extended Data Fig 11 to show lysosomal membrane tethering compatibility. We added a sentence in the end of the result session:

“.....with compatible anchoring geometry on the lysosomal membrane (Extended Data Fig. 11).”

Methods

L320: Typo CHPAS

The typo is now corrected

L341: Tagging of Raptor and mLST8: Could the authors more clearly specify which subunits were tagged at what ends ? Does tagging have any effect on the interactions discussed here as compared to pulled-down wild-type complexes ? Are any ordered tags or differences to other mTORC1 structural data observed ?

We have changed the sentence to be clear about the tagging procedure:

“The mTOR gene was cloned into a pCAG vector without a tag, the Raptor gene was cloned into a pCAG vector with an uncleavable tandem 2×Strep II-1×FLAG-tag at N-terminus, and the mLST8 gene was also cloned into a pCAG vector with an uncleavable tandem 2×Strep II-1×FLAG-tag at N-terminus.”

The tags are far away from the interaction interfaces discussed here. The co-IP and IF experiments done in the manuscript were with wild-type Raptor and mLST8, and yet showed the effects of mutants. We did not observe any ordered tags in our structures.

L347: Please provide buffer/gradient conditions for AEX.

We have now provided the buffer/gradient condition for AEX.

“The elution was diluted into equal volume of salt-free buffer (50 mM HEPES, 1mM TCEP, pH 7.4) and applied to a 5 ml HiTrap Q column (GE Healthcare). The mTORC1 complex and free RAPTOR were separated by 100ml salt gradient with salt-free buffer and high salt buffer (50 mM HEPES, 1 M NaCl, 1mM TCEP, pH 7.4).”

Fig. 4c: There is a strong reduction in total TFEB for the double and triple mutants and changes in phosphorylation aren't easily apparent by eye. Please quantify blot with pTFEB relative to TFEb and 4EBP1 or similar standards, and for change in total TFEB.

As discussed also in response to rev. 1 minor point 1, anti-pTFEB does not give results for endogenously expressed TFEB that are suitable to quantitation. Thus, phosphorylation of endogenous TFEB can only be visualized by analyzing the shift in its molecular weight. This method is a widely accepted, qualitative measure of TFEB phosphorylation and has been used

in numerous TFEB studies (e.g. Settembre et al 2012; Rocznik-Ferguson et al. 2012; Martina et al. 2012; Napolitano et al. 2020). As shown in Fig. 4c, such a shift (i.e. phosphorylation) in endogenous TFEB is markedly visible in RagC-KO cells reconstituted with wt RagC, whereas it is drastically impaired in cells expressing RagC mutants. These data are consistent with immunofluorescence data provided in Fig 4e, showing that only wt, but not mutant, RagC rescues TFEB cytosolic localization - another established readout of TFEB phosphorylation status (Settembre et al 2012; Rocznik-Ferguson et al. 2012; Martina et al. 2012; Napolitano et al. 2020).

Concerning TFEB expression, changes of endogenous TFEB levels are already known to depend on RagC expression/activation status (Napolitano et al. 2020) and are likely due to TFEB stabilization induced by Rag GTPase interaction and/or TFEB phosphorylation. However, deciphering the mechanisms that control TFEB protein stability in these conditions would require a considerable amount of work that we believe is beyond the scope of the current study.

Fig. 5: Consider to indicate membrane tethering of Ragulator for the overall megacomplex model (see also above).

We have added an Extended Data Fig 11 to show membrane tethering of the megacomplex.

Supplementary data

Table S1: More ligands than the bound GTP/GDP were observed, please mention in text and show density if relevant. B factors for protein and even more so for ligand are very low for the TFEB-Rag-Ragulator complex at around 3Å resolution, may well be fine, but please check.

Other ligands are inositol hexakisphosphate (IP6) bound to mTOR, which has been reported previously (Scaiola *et al.*, 2020) and is not the focus of this paper. We checked B factors and they are normal. The local resolution of the Raptor-TFEB-Rag-Ragulator is higher than 3 Å. We also updated the Table1, as there was one residue wrong in the Raptor model during model deposition. It doesn't affect any figure or conclusions drawn in this manuscript.

Ext. Data Fig. 8: see comments above on symmetry application and local refinement.

See the response for L216.

Referee #3 (Remarks to the Author):

We thank the reviewer for their appreciation of the manuscript's broad appeal. It is kind of them to refer to the findings as a "breakthrough". We appreciate the thoughtful questions raised by the reviewer.

Major comments:

1. The author's initial goal was to determine the structure of TFEB bound Rag-Ragulator complex. They succeeded in reconstituting a stoichiometric complex, suggesting that the interaction between TFEB and Rag-Ragulator is relatively stable. Yet, they could not resolve the density of TFEB in the resulting 3D EM density map. However, because very little information is provided in Extended data Fig 1 regarding the data processing scheme used, it is hard to determine if the inability to visualize TFEB is due solely to the dissociation of TFEB during vitrification or stringent criteria used in image processing. Furthermore, it is not clear if the authors have attempted to use approaches such as chemical crosslinking to stabilize this assembly for vitrification.

We continued the data processing of the TFEB-Rag-Ragulator, with further classification without alignment using TFEB mask. In the figure below, the TFEB mask is colored in gray, and Rag-Ragulator density is shown as a reference for the position of TFEB. The densities after 3D classification result are colored in light blue. A major class (class1) with 91.81% particles was resolved with fragmented density. Although class4 with only 0.15% particles showed some density in between RagA and RagC, we can't draw a definitive conclusion that it is TFEB due to the limited number of particles in this class.

Fig R3. 3D classification of TFEB-Rags-Ragulator without alignment

The TFEB mask is shown in gray density. The classification results are shown in blue. The Rags-Ragulator are in light gray and shown as reference to identify TFEB.

However, it is also possible that TFEB did not disassociate from Rags, but rather flexible without mTORC1. We changed the sentence of "... indicating that additional interactions were required to generate a complex stable enough for vitrification" to

"...indicating that additional interactions were required to stabilize TFEB for structural studies."

We did not attempt chemical crosslinking to stabilize the complex as later we were able to visualize TFEB when in complex with Raptor.

2. This research group has extensive experience in using HDX-MS to map interaction interfaces between subunits within protein complexes. Given that they were able to produce different complexes (eg. Rag-Ragulator) and individual subunits (eg. Raptor), HDX-MS could provide additional data to validate the interfaces observed from the cryo-EM structures.

Our lab typically uses HDX-MS in conjunction with low resolution EM studies of highly dynamic entities, when the EM itself does not provide atomistic detail. Publication-quality HDX-MS is time and resource-intensive. For a complex of this size, hundreds of peptides are generated, and the analysis is a major effort. In instances when atomistic or near-atomistic cryo-EM structure determination is possible, we typically do not carry out HDX-MS as the gain in information relative to the cost and time invested do not represent a good rate of return and we therefore did not prioritize this set of experiments. We feel that the most physiologically relevant validation consists of the functional assays of TFEB localization as carried out in cells.

3. The observation that two copies of Rag-Ragulator are present in the TFEB-Raptor-Rag-Ragulator is interesting and a major discovery of this work. It would be important to describe whether or not the canonical and non-canonical Rag-Ragulator in this assembly adopt similar structures.

We have added an Extended Data Fig.5 to show the structural comparison between canonical and non-canonical Rag-Ragulator. We added the following sentence in the manuscript:

"In addition, little structural variation was observed between c-Rag-Ragulator and nc-Rag-Ragulator (Extended Data Fig. 5)"

4. Although the presence of Raptor enabled the authors to successfully visualize TFEB interaction with Rag-Ragulator by cryo-EM and the overall structure of this complex largely resembles the one observed in the context of the megacomplex with mTORC1, this finding leads to the open question regarding if the observed TFEB-Raptor-Rag-Ragulator is a physiologically relevant intermediate. Especially given the stability of the TFEB-Raptor-Rag-Ragulator assembly, one could envisage that TFEB loading would involve first formation of the TFEB-Raptor-Rag-Ragulator prior to generation of the megacomplex with mTOR and mLST8.

It is an intriguing question whether there is an ordered mechanism for assembly of the complex. This could possibly be addressed by varying the concentrations of components and using an enzyme kinetics approach or using single molecule methods. We are not currently set up to do these experiments with the precision required to obtain conclusive results. The conclusions of this study do not depend on knowing the order of assembly and we would propose that this good question would best be resolved in the future by a lab specializing in single molecule methods and/or enzyme kinetics.

5. The authors assessed the effects of structure-guided mutants through examining the localization of TFEB as well as phosphorylation of S6K, 4EBP-1, and TFEB. It would also be interested to assess the effects of these mutants using standard autophagy assays. Related to this, have the authors generated structure-guided mutants of Raptor and assess the effects of these mutations on TFEB phosphorylation and localization?

Settembre et al (Science, 2011) previously showed that TFEB activation upregulates autophagy and an extensive literature followed after this first study supporting the role of TFEB as a master regulator of autophagy, which addresses the first question raised by the reviewer. With respect to the second, there is a difficulty with Raptor in that it is essential for cell growth and thus cannot be knocked out, making rescue experiments in cells unfeasible.

6. Rag-Ragulator plays a key role in localizing mTORC1 to the lysosomal membrane. However, there was no discussion on the effect of the second copy of Rag-Ragulator on mTORC1 membrane localization. Previous cryo-EM studies have generated a structural model regarding how mTORC1 is orientated when bound to Rag-Ragulator (Rogala et al. 2019; Anandapadamanaban et al. 2019). In this study, the observation that TFEB requires two copies of Rag-Ragulator to be recruited to mTORC1 creates a conundrum. More specifically, based on in the orientation shown in Figure 5C, the canonical Rag-Ragulator would be positioned right on top of a membrane but the non-canonical Rag-Ragulator is projected in an opposite direction away from the membrane. Can the authors explain this confusing observation?

We have included an Extended Data Fig 11 to show that the N-termini of Lamtor1 in canonical Rag-Ragulator and non-canonical Rag-Ragulator are in fact pointing to the same direction, which is compatible with lysosomal membrane tethering. We added a sentence in the end of the result session:

“.....with compatible anchoring geometry on the lysosomal membrane (Extended Data Fig. 11).”

Reviewer Reports on the First Revision:

Referees' comments:

Referee #1 (Remarks to the Author):

The authors provided reasonable explanations and revised data for my questions and comments. The additional information in this revised manuscript has further strengthened the model where distinct mTORC1 complex can be formed by TFEB and is necessary for TFEB phosphorylation. Although characterization, visualization, and process of formation of this mega-mTORC1-TFEB complex under physiological and pathological conditions remain largely elusive, the identification of the new mega-mTORC1 complex is significant and may set new branches of research to further understanding of the dynamics of mTORC1 on the lysosomal membrane. The story became mature with convincing data, and the topic is timely; thus, this reviewer supports this study for publication.

Referee #2 (Remarks to the Author):

The authors have provided a comprehensive reply to reviewers' comments and overall address the issues raised by this reviewer in an appropriate manner.

Regarding reply to L97/phosphorylation at other sites than S211 in the S211A mutant:

The authors point to published information and provide own data based on Western Blot that S142 can still be phosphorylated by mTOR, which is assuring. Ideally, samples from pull-downs and in vitro assays would be analyzed by standard phosphor-proteomic MS analysis to detect phosphorylation also at other reported mTOR phosphorylation sites.

Regarding Fig R2:

I am not quite sure what this is showing. The authors state that TFEB-WT couldn't be co-expressed and purified with active RagA/C. The SDS-PAGE for TFEB-WT has no labelled band for TFEB-WT (where would this be – overlaid with GST- RagC ?), but RagA/C are still present in similar amounts, which would argue against general toxicity (depending on experimental details) ? What are the two extra upper bands at 100 and 150kDa for TFEB-WT?

Regarding L102/absence of TFEB density of Rag-Ragulator complex:

Apparently, the authors have taken all steps in cryoEM data processing to detect even substoichiometric amounts of a bound TFEB - the results are puzzling. While I can see how a covalently linked domain of a multidomain protein may be fully disordered in an EM reconstruction, it is hard to explain why a separate subunit would bind strongly and stably without any visualized local contact. Here, in addition, we assume, as I understand, that the interaction would likely be similar as in the larger complexes where it was visualized, and binding in a "normal" manner was observed. While this part may not be critical for all the conclusions based on other structures presented here, I strongly recommend to consider all possible sources of error e.g. related to sample composition. Is all this based on a single data collection, or have samples been prepared at least

twice independently with same results ? If this brings up no other explanation, I recommend to include Figure R3 as Extended data with the manuscript.

Regarding L178/L269: Ruling out an interaction between TFEB and Raptor in the absence of other partners:

The authors note that finding a substantial (1280A²) direct interface between Raptor and TFEB was a surprise. To me it is not clear why this interface with many clearly resolved inter-residue interactions wouldn't be stable on its own. Explanations for this might be relevant and interesting and could include cooperative effects of Rag and Raptor binding. The authors quote numerous earlier co-IP experiments from cells, which is very helpful, but doesn't exclude indirect (and possibly also interesting) contributions. The most direct experiment would be to test binding with purified components, a TFEB variant and Raptor, e.g. in a simple bead-based pull-down. I would consider this particularly interesting and relevant also because of the previous and later points: The authors observe binding of TFEB to Rag-Ragulator in the absence of Raptor, but without being able to visualize TFEB, and they don't observe binding to Raptor, although it provides the largest part of the interface in the Rag-Ragulator-Raptor complex. Mutations to central residues in the Raptor-TFEB interface have no effect on function (see below/L269) and the authors conclude in their response that (the Raptor interface) is secondary to the contacts with the Rags in its contribution to function", although in the presented structures it is a highly prominent feature.

Reg. Fig. 4c: Blot quantification / shift in TFEB electrophoretic mobility as qualitative measure of phosphorylation. I still consider it useful to make an attempt at quantification of total TFEB and the shifted fraction (using boxed areas ?) simply because it is extremely difficult to guess by eye e.g. if the percentage of shifted protein is lower for the double or triple RagC mutant compared to the single mutant. The main difference is in overall TFEB concentration which is apparently going down massively for the more drastic double- or triple RagC mutants.

Notes on replies to comments of other reviewers:

Reviewer 1 asked for a quantification of the proportion of the canonical and non-canonical complexes at the lysosome, and suggested lysosome purification, mass analyses and SEC as approaches. I agree with the authors that these transient complexes with overlapping subunit composition and further cellular complexation partners could hardly be quantified in a meaningful way by the suggested approaches. A relevant read-out would probably require a highly elaborate in-cell distinction of different complex types based on specific antibody detection in fixed cells or highly localized dual-labelling for FRET etc. Establishing such tools could be a high-level scientific project in itself.

Reply to reviewer 1 – Point 4. Biochemical assays for Fig. 4g: The authors observe an apparent contradiction between Fig. 4f, substantial loss of RagC-TFEB co-localization for RagC single and double mutants, and Fig 4d, no loss of co-IP efficiency for single and double mutants of RagC. I don't fully understand why one type of interaction would be sufficiently stable for a co-IP, but additional interactions should be required for co-localization. Wouldn't the (apparently co-IP stable) nc complex alone be localized to the lysosome by virtue of the ragulator interaction? Together with the

comments on Fig. 4c, and the open points on the relevance and visualization of raptor-only and RagA/C-Ragulator complex only interactions (see above), this may warrant further attention.

Further notes:

The authors might want to consider recently (Sep 12) published papers in Nat Cell Biol, in particular Gollwitzer et al., in their discussion.

Referee #3 (Remarks to the Author):

While it would be nicer if the authors could provide more data to substantiate the observed protein-protein interfaces, the authors have explained the technical challenges associated with the suggested experiments. As such, the authors have addressed the majority of concerns raised in my report. The remaining potential issue with the manuscript is the physiological relevance of the observed TFEB-Raptor-Rag-Ragulator structure, a concern that was also raised by Reviewer 1. In the same vein, it was puzzling to read the authors' response to Reviewer's comments 1 to 3. In particular, they believed that, despite being quite stable when reconstituted in vitro, the Rag-Regulator-TFEB and the mega complex with mTORC1 that they structurally characterized are not stable assemblies under normal physiological conditions. Perhaps what the authors need to do is to clearly state this in the manuscript and comment on the limitations of their structural studies. Something like this? the manner in which TFEB is being recruited to mTORC1 is probably recapitulated in this experimental setup but spatial and temporal details on this substrate recruit mechanism cannot be delineated as the protein concentration used for reconstitution is not at physiological level.

Author Rebuttals to First Revision:

Referees' comments:

Referee #1 (Remarks to the Author):

The authors provided reasonable explanations and revised data for my questions and comments.

The additional information in this revised manuscript has further strengthened the model where distinct mTORC1 complex can be formed by TFEB and is necessary for TFEB phosphorylation.

Although characterization, visualization, and process of formation of this mega-mTORC1-TFEB complex under physiological and pathological conditions remain largely elusive, the identification of the new mega-mTORC1 complex is significant and may set new branches of research to further understanding of the dynamics of mTORC1 on the lysosomal membrane.

The story became mature with convincing data, and the topic is timely; thus, this reviewer supports this study for publication.

Thank you.

Referee #2 (Remarks to the Author):

The authors have provided a comprehensive reply to reviewers' comments and overall address the issues raised by this reviewer in an appropriate manner.

Thank you.

Regarding reply to L97/phosphorylation at other sites than S211 in the S211A mutant: The authors point to published information and provide own data based on Western Blot that S142 can still be phosphorylated by mTOR, which is assuring. Ideally, samples from pull-downs and in vitro assays would be analyzed by standard phosphor-proteomic MS analysis to detect phosphorylation also at other reported mTOR phosphorylation sites.

We have previously shown that TFEB undergoes a complex, multi-site phosphorylation mechanism that controls its subcellular localization and activity. With the exception of S211 and S142, the functional relevance of the other phosphorylation sites remains unclear (see Puertollano *et al. EMBO J.* 2018 37:e98804 for an extensive review on the topic). Thus, we believe that an MS analysis of other previously reported sites would not add meaningful content, as we do not draw any conclusions regarding other phosphorylation sites not specifically mentioned.

Regarding Fig R2:

I am not quite sure what this is showing. The authors state that TFEB-WT couldn't be co-expressed and purified with active RagA/C. The SDS-PAGE for TFEB-WT has no

labelled band for TFEB-WT (where would this be – overlaid with GST- RagC ?), but RagA/C are still present in similar amounts, which would argue against general toxicity (depending on experimental details) ? What are the two extra upper bands at 100 and 150kDa for TFEB-WT?

With respect to the query concerning co-expression, we attempted WT-TFEB-GFP co-expression with active Rags. The purification was done using glutathione resin to pull-down GST-RagC, however, WT-TFEB-GFP did not co-elute with Rags, in contrast to S211A/NLS-TFEB-GFP. The purpose of Fig. R2 was to explain why the suggestion to carry out in vitro binding of TFEB to Raptor was not feasible.

We believe WT TFEB is probably dissociated from the Rags and therefore cannot be purified using the same methodology as the rest of our study, in which S211A TFEB was purified as a complex with the Rags. While there have been a few reports from other labs of purification of isolated full-length WT TFEB, these reports did not provide evidence of mono-dispersity, and in our hands we have found this material is polydisperse and therefore reliable conclusions cannot be drawn.

We have not attempted to identify the upper bands, however, their sizes do not match TFEB and we believe they are contaminants.

Regarding L102/absence of TFEB density of Rag-Ragulator complex: Apparently, the authors have taken all steps in cryoEM data processing to detect even substoichiometric amounts of a bound TFEB - the results are puzzling. While I can see how a covalently linked domain of a multidomain protein may be fully disordered in an EM reconstruction, it is hard to explain why a separate subunit would bind strongly and stably without any visualized local contact. Here, in addition, we assume, as I understand, that the interaction would likely be similar as in the larger complexes where it was visualized, and binding in a “normal” manner was observed. While this part may not be critical for all the conclusions based on other structures presented here, I strongly recommend to consider all possible sources of error e.g. related to sample composition. Is all this based on a single data collection, or have samples been prepared at least twice independently with same results ? If this brings up no other explanation, I recommend to include Figure R3 as Extended data with the manuscript.

We have added Fig. R3 to the manuscript as Extended Data Fig. 1g.

Regarding L178/L269: Ruling out an interaction between TFEB and Raptor in the absence of other partners:

The authors note that finding a substantial (1280Å²) direct interface between Raptor and TFEB was a surprise. To me it is not clear why this interface with many clearly resolved inter-residue interactions wouldn't be stable on its own. Explanations for this might be relevant and interesting and could include cooperative effects of Rag and Raptor binding. The authors quote numerous earlier co-IP experiments from cells, which

is very helpful, but doesn't exclude indirect (and possibly also interesting) contributions. The most direct experiment would be to test binding with purified components, a TFEB variant and Raptor, e.g. in a simple bead-based pull-down. I would consider this particularly interesting and relevant also because of the previous and later points: The authors observe binding of TFEB to Rag-Ragulator in the absence of Raptor, but without being able to visualize TFEB, and they don't observe binding to Raptor, although it provides the largest part of the interface in the Rag-Ragulator-Raptor complex. Mutations to central residues in the Raptor-TFEB interface have no effect on function(see below/L269) and the authors conclude in their response that (the Raptor interface) is secondary to the contacts with the Rags in its contribution to function", although in the presented structures it is a highly prominent feature.

We purify TFEB in complex with the Rags. The Rags act in effect as chaperones that make it possible to purify non-aggregated TFEB. As described above in response to the query about Fig. R2, we have been unable so far to make isolated TFEB in non-aggregated form, so it is not feasible to do the proposed Raptor-TFEB binding study. In any event, the expectation on the basis of numerous past co-IP experiments is there would be no interaction even if it were possible to do this experiment.

Accordingly, our data, together with previously published evidence (e.g. Martina et al. & Puertollano, 2013), suggest that TFEB directly binds to the Rag GTPases and that such interaction is required for TFEB-Raptor binding. Supporting this, we have previously shown that TFEB is unable to bind to Raptor in RagA/B-KO cells, whereas reconstitution of these cells with RagA completely rescues TFEB-Raptor interaction. Although we agree with the reviewer that the TFEB-Raptor binding interface is a prominent feature of our structure, altogether our data suggest that this interaction is secondary to TFEB-Rag binding and may have a role in complex stabilization. As suggested by the reviewer, we have now added this sentence in the discussion (pg. 7): "Despite that the structures show that the Raptor-TFEB binding interface is a prominent feature, the TPAI mutational data and previously published evidence suggest that TFEB-Raptor interaction is secondary to the contacts with the Rags and may have an auxiliary role in complex stabilization."

Reg. Fig. 4c: Blot quantification / shift in TFEB electrophoretic mobility as qualitative measure of phosphorylation. I still consider it useful to make an attempt at quantification of total TFEB and the shifted fraction (using boxed areas ?) simply because it is extremely difficult to guess by eye e.g. if the percentage of shifted protein is lower for the double or triple RagC mutant compared to the single mutant. The main difference is in overall TFEB concentration which is apparently going down massively for the more drastic double- or triple RagC mutants.

This blot has been quantitated as suggested and the results shown below the blot in the revised Fig. 4c. The quantitation is consistent with the interpretation that was previously made on the basis of visual inspection.

Notes on replies to comments of other reviewers:

Reviewer 1 asked for a quantification of the proportion of the canonical and non-canonical complexes at the lysosome, and suggested lysosome purification, mass analyses and SEC as approaches. I agree with the authors that these transient complexes with overlapping subunit composition and further cellular complexation partners could hardly be quantified in a meaningful way by the suggested approaches. A relevant read-out would probably require a highly elaborate in-cell distinction of different complex types based on specific antibody detection in fixed cells or highly localized dual-labelling for FRET etc. Establishing such tools could be a high-level scientific project in itself.

Thank you.

Reply to reviewer 1 – Point 4. Biochemical assays for Fig. 4g: The authors observe an apparent contradiction between Fig. 4f, substantial loss of RagC-TFEB co-localization for RagC single and double mutants, and Fig 4d, no loss of co-IP efficiency for single and double mutants of RagC. I don't fully understand why one type of interaction would be sufficiently stable for a co-IP, but additional interactions should be required for co-localization. Wouldn't the (apparently co-IP stable) nc complex alone be localized to the lysosome by virtue of the ragulator interaction? Together with the comments on Fig. 4c, and the open points on the relevance and visualization of raptor-only and RagA/C-Ragulator complex only interactions (see above), this may warrant further attention.

It is expected to observe slightly different results with the use of different experimental approaches. There may be differences in specificity and stringency between immunofluorescence and co-IPs. In the mentioned single and double mutations of RagC, we observe loss of lysosomal localization and phosphorylation of TFEB albeit the detection of TFEB in the RagC IPs. The results raise the possibility that the mutations introduced alter the residence time of the nc-Rags but not c-Rags on lysosomes, because the interaction between Rags and Ragulator is not static in nature (Rosalie et.al PMID:30061680). The co-IP results confirm that the megacomplex can still be formed in bulk, but the IF results suggest that significant recruitment of TFEB at any given time point is low due to decreased affinity for ragulator, which in turn effectively prevents TFEB phosphorylation.

Please note that this comment was originally from reviewer 1, who is now satisfied.

Further notes:

The authors might want to consider recently (Sep 12) published papers in Nat Cell Biol,

in particular Gollwitzer et al., in their discussion.

The manuscript by Gollwitzer et al. suggests that RagC and RagD may differentially modulate mTORC1 activity towards its substrates TFEB and S6K. In particular, they suggest that RagC plays a minor role in TFEB phosphorylation.

Our data, in line with robust previously published evidence obtained by several groups including our own, clearly show that RagC plays a crucial role in TFEB phosphorylation for the following reasons:

1. TFEB forms a stable complex with a Rag GTPase dimer composed of RagA and RagC (present manuscript and PMID: 32612235, 23524842)
2. TFEB and TFE3 are totally de-phosphorylated and nuclear in RagC-KO cells, despite the presence of endogenous RagD. Reconstitution of these cells with wt or active RagC is sufficient to rescue TFEB/TFE3 phosphorylation and subcellular localization (present manuscript and PMID: 35358174, 34253722)
3. The expression of a constitutively active form of RagC is sufficient to rescue TFEB/TFE3 phosphorylation and subcellular localization in FLCN-KO cells (PMID: 32612235, 35358174)

We have now added content in the discussion to better clarify these points.

“All of the TFEB-contacting residues of RagC are conserved in RagD, consistent with the finding that expression of either RagC or RagD can rescue TFEB recruitment to lysosomes in RagC/D double knockout (KO) cells ⁷. However, a recent publication {Gollwitzer, 2022 #131} suggested that RagC only plays a minor role in the recruitment and phosphorylation of TFEB. In contrast with this conclusion, here we showed that TFEB forms a stable complex with a RagA-RagC dimer, and that TFEB phosphorylation and subcellular localization are drastically affected in RagC KO cells, despite the presence of endogenous RagD. These results are in line with previous publications showing that RagC depletion promotes TFEB/TFE3 de-phosphorylation and nuclear translocation {Alesi, 2021 #103; Li, 2022 #132} and that the expression of a constitutively active RagC mutant is sufficient to rescue TFEB/TFE3 phosphorylation and subcellular localization in FLCN KO cells {Napolitano, 2020 #12; Li, 2022 #132}. Together, these data clearly demonstrate that RagC plays a major role in TFEB phosphorylation.

Referee #3 (Remarks to the Author):

In particular, they believed that, despite being quite stable when reconstituted in vitro, the Rag-Regulator-TFEB and the mega complex with mTORC1 that they structurally characterized are not stable assemblies under normal physiological conditions. Perhaps what the authors need to do is to clearly state this in the manuscript and comment on the limitations of their structural studies. Something like this? the manner in which TFEB is being recruited to mTORC1 is probably recapitulated in this experimental setup but spatial and temporal details on this substrate recruit mechanism cannot be delineated as the protein concentration used for reconstitution is not at physiological level.

In response to this point, new text reading " The question of how the megacomplex dissociates in cells following Ser211 phosphorylation remains open. Dissociation might be mediated by phosphorylation-triggered structural rearrangements within TFEB or other megacomplex components, or binding to 14-3-3 proteins²⁻⁵ or other additional factors." added to end of para. 2 in the discussion.

Reviewer Reports on the Second Revision:

Referees' comments:

Referee #2 (Remarks to the Author):

The authors have addressed all reviewer's comments in their updated manuscript and the rebuttal letter, this has helped to clarify open points. The manuscript is ready for publication, I just suggest to make sure that some information from the review process is represented in the manuscript.

REPLY TO "REGARDING FIG 2":

The authors have altered their explanation for why certain experiments with wtTFEB were not possible. I recommend that the Fig. R2 from the first rebuttal letter should be published with the manuscript in any place together with a clarifying statement in the methods, based on the statement from below: "however, WT-TFEB-GFP did not co-elute with Rags, in contrast to S211A/NLS-TFEB-GFP"

(

Revision 2: With respect to the query concerning co-expression, we attempted WT-TFEB-GFP coexpression with active Rags. The purification was done using glutathione resin to pulldown GST-RagC, however, WT-TFEB-GFP did not co-elute with Rags, in contrast to S211A/NLS-TFEB-GFP. The purpose of Fig. R2 was to explain why the suggestion to carry out in vitro binding of TFEB to Raptor was not feasible. We believe WT TFEB is probably dissociated from the Rags and therefore cannot be purified using the same methodology as the rest of our study, in which S211A TFEB was purified as a complex with the Rags. While there have been a few reports from other labs of purification of isolated full-length WT TFEB, these reports did not provide evidence of mono-dispersity, and in our hands we have found this material is polydisperse and therefore reliable conclusions cannot be drawn.

Revision 1: As for wild type TFEB, we couldn't co-express and purify it with active RagA/C in HEK293F cells, presumably due to its toxicity when highly over-expressed.

)

REGARDING REPLY TO REVIEWER 1 – POINT 4:

I suggest to include/represent a statement (with modifications as needed) from the rebuttal letter in the manuscript, as it very clearly summarizes this point: "The co-IP results confirm that the megacomplex can still be formed in bulk, but the IF results suggest that significant recruitment of TFEB at any given time point is low due to decreased affinity for regulator, which in turn effectively prevents TFEB phosphorylation"

SUPPL. DATA FOR FIG 3C:

Please check. The top row shows multiple captures of the same blot but all indicated bands are from the variant in the middle.

Author Rebuttals to Second Revision:

Referees' comments:

Referee #2 (Remarks to the Author):

The authors have addressed all reviewer's comments in their updated manuscript and the rebuttal letter, this has helped to clarify open points. The manuscript is ready for publication, I just suggest to make sure that some information from the review process is represented in the manuscript.

Thank you.

REPLY TO "REGARDING FIG 2":

The authors have altered their explanation for why certain experiments with wtTFEB were not possible. I recommend that the Fig. R2 from the first rebuttal letter should be published with the manuscript in any place together with a clarifying statement in the methods, based on the statement from below: "however, WT-TFEB-GFP did not co-elute with Rags, in contrast to S211A/NLS-TFEB-GFP"

This rebuttal figure has been added to Supplementary Fig. 2 and cited near the end of the methods section on protein expression and purification.

REGARDING REPLY TO REVIEWER 1 – POINT 4:

I suggest to include/represent a statement (with modifications as needed) from the rebuttal letter in the manuscript, as it very clearly summarizes this point: "The co-IP results confirm that the megacomplex can still be formed in bulk, but the IF results suggest that significant recruitment of TFEB at any given time point is low due to decreased affinity for ragulator, which in turn effectively prevents TFEB phosphorylation"

This statement was added in the results in the sentence following where Fig. 4d is cited.

SUPPL. DATA FOR FIG 3C:

Please check. The top row shows multiple captures of the same blot but all indicated bands are from the variant in the middle.

The unused duplicate exposures were removed.

Editorial points

1. *Please ensure that the text size in all figures is at least 5 pt Arial.*

Done

2. *Please reduce the article title to 75 characters (with spaces) or less.*

Done

3. *Please ensure that you describe each figure panel in the figure legends.*

Done

4. Please ensure all main figure legends are 300 words or less.

Done

5. Figures 1, 4 and 5 are too tall in height when re-sized to 18 cm width, please reduce to 17 cm or less.

Done

6. Please reduce subheadings to 40 characters (with spaces) or less.

This has been done.

7. Please provide a supplementary information guide (see below)

The SI Guide file has been added.

8. Supplementary videos currently do not have titles/legends, please provide these.

Done, in the SI Guide file

9. Competing interests for only a few authors have been declared, (J.H.H.; A.B.); not for all the authors. Please provide a statement that encompasses all authors (i.e. add "... the rest of the authors declare no conflicts of interest.")

Done

10. The manuscript includes the following PDB/EMDB accession codes, which are yet to be released. PDB (7UX2, 7UXC, 7UXH) and EMDB (26840, 26842, 26843, 26844, 26846, 26852, 26857, 26861). Please ensure that these are made public ASAP.

We have emailed the RCSB and EMDB to release coordinates and density as soon as possible.

11. In the 'Data presentation' section of Editorial Policy Checklist, the fields "Individual data points are shown when possible, and always for $n \leq 10$ " and the one corresponding to data distribution format were marked as N/A; however, it is observed that both fields are relevant to the study. Please revise the checklist accordingly.

Done

12. In the Editorial Policy Checklist under "Additional policy considerations", the provision of an official PDB validation report was confirmed; however, no such document was observed among the manuscript files. Please either add this (not strictly required) or revise accordingly.

We have uploaded a combined pdf of all of the validation reports.

13. We noticed that the full scans of all gel images have been bordered with a solid box-border, we ask authors to instead use a light-dotted line. Please revise accordingly.

We have changed the box borders as requested.

14. Figure 3h appears to be mislabeled as 'l' in the corresponding figure legend. This needs to be rectified.

Done

15. Please note that the full scans for extended data figures 5a, 5b, 7e, 8c are mislabeled as extended data figures 6a, 6b, 8, 9c respectively. This needs to be rectified.

We have corrected this.